# Vascular Endothelial Cell-Derived Exosomal Sphingosylphosphorylcholine Attenuates Myocardial Ischemia–Reperfusion Injury through NR4A2-Mediated Mitophagy

**DOI:** 10.3390/ijms25063305

**Published:** 2024-03-14

**Authors:** Yifan Yu, Zhiliang Li, Yuqing Cai, Jiahui Guo, Yushuang Lin, Jing Zhao

**Affiliations:** Shandong Provincial Key Laboratory of Animal Cells and Developmental Biology, School of Life Sciences, Shandong University, Qingdao 266237, China; 15265213775@163.com (Y.Y.); lzl293451@163.com (Z.L.); 15835112275@163.com (Y.C.); guojiahui21@163.com (J.G.); linyushuang@sdu.edu.cn (Y.L.)

**Keywords:** exosome, sphingosylphosphorylcholine, mitophagy, NR4A2, optineurin

## Abstract

Cardiomyocyte survival is a critical contributing process of host adaptive responses to cardiovascular diseases (CVD). Cells of the cardiovascular endothelium have recently been reported to promote cardiomyocyte survival through exosome-loading cargos. Sphingosylphosphorylcholine (SPC), an intermediate metabolite of sphingolipids, mediates protection against myocardial infarction (MI). Nevertheless, the mechanism of SPC delivery by vascular endothelial cell (VEC)-derived exosomes (VEC-Exos) remains uncharacterized at the time of this writing. The present study utilized a mice model of ischemia/reperfusion (I/R) to demonstrate that the administration of exosomes via tail vein injection significantly diminished the severity of I/R-induced cardiac damage and prevented apoptosis of cardiomyocytes. Moreover, SPC was here identified as the primary mediator of the observed protective effects of VEC-Exos. In addition, within this investigation, in vitro experiments using cardiomyocytes showed that SPC counteracted myocardial I/R injury by activating the Parkin and nuclear receptor subfamily group A member 2/optineurin (NR4A2/OPTN) pathways, in turn resulting in increased levels of mitophagy within I/R-affected myocardium. The present study highlights the potential therapeutic effects of SPC-rich exosomes secreted by VECs on alleviating I/R-induced apoptosis in cardiomyocytes, thereby providing strong experimental evidence to support the application of SPC as a potential therapeutic target in the prevention and treatment of myocardial infarction.

## 1. Introduction

Acute myocardial infarction, a prevalent ischemic heart disease, is associated with high morbidity and mortality risks worldwide [1]. The major limiting factor for the treatment of acute myocardial infarction is myocardial ischemia/reperfusion (I/R) injury, which is closely associated with the progressive loss of cardiomyocytes [2]. Rapid advances in basic research and clinical treatments have not contributed to the development of strategies to treat or effectively delay the progression of ischemic heart diseases [3]. Recent studies have reported the protective effects of ultramicronized palmitoylethanolamide (PEA-um) and a mixture of PEA-um and baicalein on myocardial I/R injury in vivo, indicating that lipids are potential therapeutic agents for cardiovascular diseases [4]. Compared with dietary supplementation and intravenous injection, lipid nanoparticles and liposomes are efficient lipid delivery systems owing to their simple administration mode, controlled drug release, and limited oxidation and degradation [5,6]. For example, lipid nanoparticles have been used for vitamin E delivery to the skin [7], while liposomes are used as a spray for drug delivery to the respiratory system [8]. However, the targeting efficiency of lipids is poor. Additionally, the identification of endogenous lipids with myocardial protective effects is challenging.

Exosomes are nanosized extracellular vesicles (EVs) whose membranes are rich in sphingolipid, cholesterol, and phosphatidylcholine with di-saturated fatty acyl chains [9,10]. Exosomes are considered advantageous for the specific delivery of drugs to cells due to their nano size and characteristics of natural liposomes [11]. Previous studies on exosomes have focused on the transportation of protein and microRNA (miRNA) cargos during intercellular communication. Further studies are needed to explore the roles of exosomal lipids [12]. Exosomal lipids are involved in various cellular functions during intercellular communication [13]. For example, exosomes rich in palmitic acid esters induce the proliferation of myoblasts [14], while those rich in sphingosine-1-phosphate and prostaglandin E2 (PGE2) promote the recruitment of Th17 cells in the intestine and the subsequent establishment of tumors in the colon [15]. Exosomal PGE2 enhances the expansion of myeloid-derived suppressor cells and promotes tumor growth [16]. However, the roles of exosomal lipids in cardiomyocytes are unclear.

Sphingosylphosphorylcholine (SPC), an intermediate metabolite of sphingolipid, is widely distributed in the cell membrane and cytoplasm of various tissues and organs, including the blood, heart, blood vessels, and brain [17]. Previous studies have reported that SPC enhances the autophagy of cardiomyocytes via lipid rafts [18] and inhibits the apoptosis of hypoxic cardiomyocytes, alleviating infarction and exerting myocardial protective effects [19,20,21]. In addition, studies have shown that High-Density Lipoprotein (HDL)-associated SPC directly protects against myocardial reperfusion injury in vivo via the Sphingosine-1-phosphate (S1P)_3_ receptor [22]. However, previous studies have not examined if SPC is stored or generated in the exosomes or the role of SPC in the exosome-dependent biological activities.

Nuclear receptor subfamily group A member 2 (NR4A2), also known as NURR1, is an orphan nuclear receptor belonging to the NR4A family of receptors [23]. Previous studies have suggested that NR4A2 regulates the pathogenesis of various heart diseases, including diabetic cardiomyopathy, cardiac hypertrophy, left ventricular dilation, and heart failure by modulating the autophagy flux, cardiac metabolism, and hemodynamics [24,25,26]. However, the roles of NR4A2 in myocardial I/R injury are controversial [27,28]. Previously, we reported that SPC exerts protective effects on the ischemic myocardium by upregulating the autophagic flux through NR4A2 [29]. However, further studies are needed to examine if exosomal SPC exerts protective effects during the reperfusion phase through NR4A2.

This study demonstrated the presence of SPC in vascular endothelial cell-derived exosomes (VEC-Exos) and revealed a novel method for modulating the exosomal SPC content. Additionally, this study, for the first time, revealed that SPC in VEC-Exos significantly improves cardiac function and alleviates myocardial injury in a mice model of myocardial I/R injury. In vitro experiments using cardiomyocytes suggested that SPC promotes mitophagy in cardiomyocytes with I/R injury by inhibiting the apoptosis of cardiomyocytes through the activation of the Parkin and NR4A2/optineurin (OPTN) pathways. These results suggest that SPC in human umbilical vascular endothelial cell (HUVEC)-derived exosomes (HUVEC-Exos) is a potential therapeutic target for myocardial I/R injury.

## 2. Results

### 2.1. Isolation and Characterization of Human Umbilical Vascular Endothelial Cell-Derived Exosomes (HUVEC-Exos)

To obtain HUVEC-Exos, the exosomes in the HUVEC culture medium were purified using ultracentrifugation. Transmission electron microscopy revealed that these exosomes exhibited a disc shape (Figure 1A). Western blotting analysis revealed the expression of the characteristic cell surface antigens CD63 and TSG101 in HUVEC-Exos (Figure 1B). The average size of the exosomes was 151.4 nm, which is consistent with the range of the normal size of exosomes. (Figure 1C). These results indicated the successful extraction of HUVEC-Exos.

### 2.2. HUVEC-Exos Inhibit Myocardial Ischemia/Reperfusion (I/R) Injury

To investigate the functions of HUVEC-Exos, a mice model of myocardial I/R was established. Mice were intravenously injected with exosomes and the blood perfusion was restored at 45 min post-myocardial ischemia for 24 h. To determine the optimal exosome dosage, dose gradient experiments were performed with exosomal concentrations of 5–25 μg/10 g bodyweight. The myocardial proteins were subsequently extracted to detect the levels of the apoptotic proteins PARP1 and cleaved caspase3. At a dosage of 25 μg/10 g bodyweight, exosomes significantly mitigated the I/R-induced upregulation of PARP1 and cleaved caspase3 (Figure 2A–C). Thus, subsequent experiments were performed with exosomes at a dosage of 25 μg/10 g bodyweight. Cardiac ultrasound revealed that exosomes significantly reversed the I/R-induced changes in ejection fraction (EF) and shortening fraction (FS) (Figure 2D–F). The infarct area (the white area) in the mice heart was evaluated using tetrazolium chloride (TTC) staining. Compared with that in the sham group, the infarct area was significantly higher in the model group. However, exosome administration significantly decreased the infarct area in the mice heart (Figure 2G–H).

### 2.3. Effect of Sphingosylphosphorylcholine (SPC) Upregulation and Downregulation in HUVEC-Exos

Next, the presence of SPC in HUVEC-Exos was examined. SPC was detected in the exosomal lipid extracts (Figure 3A). At 2 h post-exosome administration, the lipid extracts of the myocardium were examined using high-performance liquid chromatography equipped with an evaporative light scattering detector (HPLC-ELSD). HUVEC-Exos upregulated the myocardial levels of SPC (Figure 3B). These findings suggested that HUVEC-Exos can transport SPC to the myocardium. To further determine if HUVEC-Exos exerted myocardial protective effects through SPC, the content of SPC in HUVEC-Exos was modulated by transfecting the cells with small interfering RNA (siRNA) targeting ATX (si-ATX) or ATX-overexpressing plasmid (pcDNA3.1-ATX). ATX can hydrolyze SPC [30]. The knockdown and overexpression efficiencies of siRNA and overexpression constructs, respectively, were validated (Appendix A). The exosomes were subjected to HPLC to determine the SPC content. ATX knockdown significantly upregulated the SPC content in HUVEC-Exos. In contrast, ATX overexpression downregulated the SPC content in HUVEC-Exos (Figure 3C). Mice were administered with si-ATX-transfected or pcDNA3.1-ATX-transfected HUVEC-Exos. The administration of si-ATX-transfected HUVEC-Exos potentiated the HUVEC-Exo-induced restoration of myocardial EF and FS in I/R mice. In contrast, the administration of pcDNA3.1-ATX-transfected HUVEC-Exos suppressed the HUVEC-Exo-induced restoration of EF and FS in I/R mice (Figure 3D–F). Consistently, TTC staining demonstrated that the administration of si-ATX-transfected HUVEC-Exos potentiated the HUVEC-Exo-induced downregulation of infarct size. In contrast, the administration of pcDNA3.1-ATX-transfected HUVEC-Exos suppressed the HUVEC-Exo-induced downregulation of infarct size (Figure 3G–H). Western blotting analysis revealed that the administration of si-ATX-transfected HUVEC-Exos potentiated the HUVEC-Exo-induced downregulation of apoptosis-related proteins, whereas that of pcDNA3.1-ATX-transfected HUVEC-Exos suppressed the HUVEC-Exo-induced downregulation of apoptosis (Figure 3I–K). These findings further suggest that HUVEC-Exos exert cardioprotective effects through SPC.

### 2.4. Exosomal SPC Inhibits Apoptosis in the Hypoxia/Reoxygenation (H/R)-Induced Cardiomyocyte Injury Model

To verify the protective effect of exosomal SPC on cardiomyocytes, H9c2 cells were used to establish an in vitro H/R model. H9c2 cells were incubated with 3,3′-dioctadecyloxacarbocyanine perchlorate (DiO)-labeled HUVEC-Exos. Laser scanning confocal microscopy revealed the internalization of HUVEC-Exos in H9c2 cells (Figure 4A). To identify the optimal exosome dosage for in vitro experiments, gradient experiments were performed with exosomes at concentrations of 1, 2, 4, and 8 μg/mL. The results of the 3-(4,5-dimethylthiazol-2-yl)-2,5-diphenyltetrazolium bromide assay revealed that the viability of cardiomyocytes significantly improved upon treatment with HUVEC-Exos at a concentration of 8 μg/mL. Thus, subsequent in vitro experiments were performed with exosomes at a concentration of 8 μg/mL (Figure 4B). Western blotting analysis demonstrated that HUVEC-Exos downregulated the level of apoptosis-related proteins in H/R-treated H9c2 cells (Figure 4C–E). Next, H/R-H9c2 cells were incubated with si-ATX-transfected or pcDNA3.1-ATX-transfected HUVEC-Exos. Western blotting analysis revealed that si-ATX-transfected HUVEC-Exos significantly potentiated the HUVEC-Exo-induced downregulation of apoptosis-related proteins in cardiomyocytes. In contrast, pcDNA3.1-ATX-transfected HUVEC-Exos suppressed the HUVEC-Exo-induced downregulation of apoptosis-related proteins (Figure 4F–H).

### 2.5. SPC Exerts Cardioprotective Effects against I/R Injury by Promoting Mitophagy through NR4A2

We have previously demonstrated that HUVEC-Exos can exert robust cardioprotective effects via SPC. Next, we aim to elucidate the molecular mechanisms by which SPC functions. Considering the complexity of exosomal contents and the potential for other substances to interfere with experimental outcomes, we will proceed to treat H9c2 cells with SPC monomers to validate the molecular mechanisms underlying SPC’s effects. Previously, we demonstrated that SPC exerts protective effects on the ischemic myocardium by upregulating autophagy through NR4A2 [29]. Mitochondria are the main sites of initial H/R injury in cardiomyocytes. Thus, this study hypothesized that SPC exerts cardioprotective effects in H/R-treated H9c2 cells by regulating mitophagy through NR4A2. A time gradient in the reoxygenation phase of H9c2 cells (1, 2, 4, and 6 h) was established to evaluate the level of NR4A2 using Western blotting analysis. The level of NR4A2 gradually decreased with the increase in reoxygenation time up to 6 h and significantly decreased thereafter (Figure 5A,B). Additionally, Western blotting analysis revealed that SPC upregulated the NR4A2 levels in H/R-treated H9c2 cells (Figure 5C,D). Immunofluorescence analysis was performed to label the mitochondria and two mitophagy markers (Parkin and LC3 II). The co-localization of these two markers with the mitochondria was examined using laser confocal microscopy. SPC promoted the co-localization of both Parkin and LC3 II with the mitochondria in H/R-treated H9c2 cells (Figure 5E–H). Next, the mitochondrial proteins were subjected to Western blotting. The mitochondrial Parkin level was significantly upregulated (Figure 5I,J). These results suggest that SPC exerts anti-apoptotic effects in H/R-treated H9c2 cells by upregulating mitophagy through NR4A2.

### 2.6. NR4A2 Knockdown Mitigates the Cardioprotective Effect of SPC

To determine the function of NR4A2, the H9c2 cells were transfected with si-*NR4A2*. The knockdown efficiency of si-*NR4A2* was verified (Appendix A). *NR4A2* knockdown mitigated the SPC-induced downregulation of apoptosis (Figure 6A–C). Additionally, *NR4A2* knockdown suppressed the SPC-induced co-localization of LC3 II with the mitochondria but did not affect the SPC-induced co-localization of Parkin with the mitochondria (Figure 6D–G). The results of the Western blotting analysis of mitochondrial proteins were consistent with those of immunofluorescence analysis (Figure 6H,I). These results suggest that *NR4A2* knockdown mitigates the protective effect of SPC on H/R-treated myocardium.

### 2.7. NR4A2 Overexpression Potentiates SPC-Mediated Protective Effects

To further clarify the function of NR4A2, the H9c2 cells were transfected with pcDNA3.1-NR4A2. The overexpression efficiency of pcDNA3.1-NR4A2 was validated (Appendix A). NR4A2 overexpression potentiated the inhibitory effect of SPC on apoptosis (Figure 7A–C). Additionally, NR4A2 overexpression potentiated the SPC-induced co-localization of LC3 II with the mitochondria but did not significantly affect the SPC-induced co-localization of Parkin with the mitochondria (Figure 7D–G). The results of the Western blotting analysis of mitochondrial proteins were consistent with those of the immunofluorescence analysis (Figure 7H,I). These results suggest that SPC exerts protective effects in H/R-treated H9c2 cells by promoting mitophagy through NR4A2.

### 2.8. NR4A2 Promotes Mitophagy through OPTN

To investigate the molecular mechanism underlying the function of NR4A2, the binding site of NR4A2 in the promoter region of OPTN (ATGTTAACCTTTTTT), which encodes one of the mitophagy receptors, was predicted using a database (www.genomatix.de/ (accessed on 27 February 2016)). The wild-type or mutant OPTN promoter was individually cloned into the dual-luciferase reporter plasmid vector PGL3. The recombinant plasmids were co-transfected with NR4A2 overexpression plasmid into 293T cells. The activation of OPTN promoter by NR4A2 was evaluated using the dual-luciferase reporter gene kit. The chemiluminescence of wild-type and mutant OPTN promoters was quantified using a microplate reader. NR4A2-mediated promoter activation was significantly impaired in the mutant OPTN-transfected group, indicating that NR4A2 regulated the transcription and translation of OPTN (Figure 8A). SPC promoted the co-localization of OPTN with the mitochondria in H/R-treated H9c2 cells (Figure 8B,C). To further confirm the association between NR4A2 and OPTN, the H9c2 cells were transfected with si-NR4A2 and pcDNA3.1-NR4A2. Laser confocal microscopy analysis revealed that NR4A2 knockdown inhibited the SPC-induced co-localization of OPTN with the mitochondria. In contrast, NR4A2 overexpression potentiated the SPC-induced co-localization of OPTN with the mitochondria (Figure 8D–G). Next, the H9c2 cells were transfected with si-OPTN or an overexpression plasmid (pCMV-OPTN). The knockdown and overexpression efficiencies of siRNA and overexpression constructs, respectively, were verified (Appendix A). OPTN knockdown suppressed the SPC-induced co-localization of LC3 II with the mitochondria. In contrast, OPTN overexpression potentiated the SPC-induced co-localization of LC3 II with the mitochondria (Figure 8H–K). These results suggest that NR4A2 exerts myocardial protective effects by promoting mitophagy through the upregulation of OPTN. The differential co-localization of OPTN with the mitochondria between the si-NR4A2-transfected and si-NR4A2 + SPC groups was analyzed. The co-localization of OPTN with the mitochondria in the si-NR4A2 + SPC group was higher than that in the si-NR4A2 group (Figure 8F,G). Thus, SPC promotes mitochondrial OPTN recruitment through the Parkin pathway.

### 2.9. SPC in HUVEC-Exos Upregulates the Expression of NR4A2, OPTN, and Parkin in the Myocardial Tissues of I/R Mice

To further verify the in vitro results, HUVEC-Exos, si-ATX-transfected HUVEC-Exos, and pcDNA3.1-ATX-transfected HUVEC-Exos were injected into the tail vein of I/R mice. The myocardial levels of NR4A2 and OPTN were examined. Western blotting analysis revealed that HUVEC-Exos significantly upregulated the myocardial levels of NR4A2 and OPTN. The mitochondria in the myocardium were isolated using a kit. Western blotting analysis demonstrated that the mitochondrial Parkin levels were significantly upregulated. Additionally, the myocardial expression levels of NR4A2, OPTN, and Mito-Parkin were further upregulated upon administration of si-ATX-transfected HUVEC-Exos. The administration of pcDNA3.1-ATX-transfected HUVEC-Exos significantly downregulated the expression levels of NR4A2, OPTN, and Mito-Parkin (Figure 9). The results of in vivo experiments were consistent with those of in vitro experiments. Thus, SPC exerts cardioprotective effects by simultaneously activating the Parkin and the NR4A2/OPTN pathways.

## 3. Discussion

Exosomes are regarded as the new participant of cell-to-cell communication in the myocardial infarct microenvironment [31]. Numerous studies have documented the varied functions of exosomes derived from diverse sources in the context of myocardial infarct. It has been shown cardiac fibroblasts were activated when incubated with miR-195-rich cardiosomes isolated from ischemic cardiomyocytes [32]. Moreover, cardiac progenitor cell-derived exosomes have been suggested to be a potent protective agent against acute I/R-induced injury in cardiac myocytes [33,34,35]. Among them, exosomes secreted by vascular endothelial cells play a significant role in the pathogenesis of cardiovascular diseases, particularly myocardial infarction, through various mechanisms. Lin et al. demonstrated that exosomes secreted by vascular endothelial cells can activate macrophages through the MAPK/NF-κB signaling pathway during atherosclerosis progression [36]. In another study, Su et al. found that exosomal LINC00174 generated from vascular endothelial cells repressed p53-mediated autophagy and apoptosis in cardiomyocytes to mitigate I/R-induced myocardial damage [37]. Recent research has increasingly concentrated on elucidating the functional roles of diverse RNA species, including miRNAs and lincRNAs, within the exosomes derived from vascular endothelial cells. However, the roles of exosomal lipids in cardiomyocytes are unclear. Sphingolipids, as integral components of exosomes, exhibit significant biological activities, and the metabolism of sphingolipids is intricately associated with the pathogenesis of myocardial infarction. [38]. Consequently, investigating the role of sphingomyelins within exosomal compositions in the context of myocardial infarction is of paramount importance. For the first time, our study identified the presence of SPC in exosomes secreted by vascular endothelial cells, and demonstrated that it exerts a considerable protective effect against myocardial I/R injury. This investigation furnishes novel evidence for the involvement of sphingolipids in exosomes from vascular endothelial cells in the modulation of myocardial function.

The underlying mechanisms of I/R are complex, including damage from ischemia and hypoxia and subsequent reperfusion injury. Current studies suggest that endothelial dysfunction, immune activation, and inflammatory response are key factors in triggering MI/RI, while reactive oxygen species (ROS), intracellular Ca^2+^ overload, and mitochondrial permeability transition pore (mPTP) opening are crucial molecular mediators [39]. It has been shown that different concentrations of SPC can induce pulmonary artery vasoconstriction by promoting Ca^2+^ input through different pathways [40,41]. In endothelial cells, SPC generates reactive oxygen species through calcium-, protein kinase Cδ-, and phospholipase D-dependent pathways, and induces apoptosis of endothelial cells through reactive oxygen species-mediated activation of ERK [42,43]. Ge et al. showed that in vascular endothelial cells, SPC inhibits apoptosis by promoting autophagy [44]. In addition, SPC was demonstrated to have a beneficial effect in a mouse model of transient myocardial I/R. It has become increasingly clear that SPC is intricately connected with multiple pivotal factors implicated in myocardial infarction. Our findings substantiated that SPC in the HUVEC-Exos markedly ameliorated the extent of myocardial I/R injury in murine models. However, the uptake of HUVEC-Exos by cardiomyocytes appeared to be non-specific. Rigorous experimental elucidation of HUVEC-Exos’ impact on blood vessels and other organ tissues post-tail vein injection in mice is imperative to comprehensively assess the role of SPC within HUVEC-Exos in myocardial infarction. Concurrently, our observations revealed that SPC exhibits multifaceted roles within identical cellular environments, a variance possibly attributable to the concentration gradients of SPC. Moreover, the present body of research consistently indicates that SPC confers a protective effect upon cardiomyocytes throughout both ischemic and reperfusion phases. It may thus be prudent to diversify the concentration range of SPC applied and scrutinize any resultant variations in its impact on cardiomyocytes, in pursuit of a thorough understanding of SPC’s regulatory influence on these cells.

ATX, a secreted enzyme with phosphodiesterase and lysophospholipase D activities, catalyzes the production of physiologically active lysophospholipids, especially lysophosphatidic acid (LPA) [45]. The ATX-LPA signaling axis is considered an important regulator of the pathogenesis of cardiovascular diseases [46]. ATX activity may be closely related to the development of cardiovascular diseases, such as atherosclerosis [47], hypertension [48], heart failure, and myocardial infarction [49]. Several experimental models and clinical studies have indicated that the inhibition of ATX activity or ATX-induced LPA signaling suppresses atherosclerotic plaque formation and decreases cardiac remodeling and fibrosis. Furthermore, ATX functions as a lyase, hydrolyzing SPC [30]. Based on these findings, this study developed an effective method to alter the SPC content in exosomes by modulating the expression of ATX. Exosomes with upregulated and downregulated SPC contents were obtained using this method. In concert with our findings, we postulate that a segment of ATX’s functionality might be mediated through the modulation of SPC hydrolysis. Elucidating the correlation between fluctuations in ATX expression and SPC concentration levels during myocardial infarction could unveil novel therapeutic avenues for this condition.

As lipid sorting and enrichment in the EV membranes of cells vary, the cell line and cell type are important factors affecting exosomal lipid production [13]. For example, ceramides are rich in exosomes secreted by N2a cells. Thus, N2a cell-derived exosomes are considered the optimal delivery tools for ceramides [50]. This study demonstrated that SPC was enriched in HUVEC-Exos and that this SPC can significantly inhibit myocardial I/R injury. In terms of drug physicochemical properties, small-molecule drugs are readily understood within the body and exhibit a brief duration of action [51]. Consequently, the investigation into the function of SPC monomers in vivo via intravenous injection may confront issues of inefficiency [17]. Exosomes possess innate material transport capabilities, inherent long-term recycling capacity, and favorable biocompatibility, among other attributes, rendering them highly promising as drug delivery vehicles [52]. Therefore, VEC-derived Exos are potential SPC carriers and can be used for the treatment of cardiac injury. However, targeting the receptor cells remains a problem [53]. Wang et al. fused exosome-enriched membrane proteins with ischemic myocardial targeting peptide (IMTP) and were able to be effectively internalized by hypoxia-injured H9c2 cells and had more accumulation in mouse ischemic myocardium compared to blank exosomes [54]. Consequently, the optimization of the affinity of HUVEC-Exos for cardiomyocytes through modification of the exosomal membrane constitutes a pivotal query to be explored in our forthcoming experimental endeavors. Simultaneously, it is imperative to conduct a head-to-head analysis comparing the efficacious doses of SPC monomers and HUVEC-Exos-SPC within the same experimental framework to elucidate the disparity in their efficacies.

Mitochondria have been recognized as a key trigger for myocardial I/R. Mitochondrial quality control mechanisms are a series of adaptive responses after I/R to preserve mitochondrial structure and function and ensure cardiomyocyte survival and cardiac function [55]. It was found that SPC has a role in regulating mitochondrial function. For instance, SPC could regulate myocardial mitochondrial homeostasis by activating the MORN4-MFN2 axis during the ischemic situation. This study demonstrated that SPC inhibited apoptosis in H/R-treated cardiomyocytes and promoted the recruitment of Parkin to the mitochondria, which can be attributed to SPC-mediated Parkin activation. Mitophagy is reported to attenuate rather than exacerbate myocardial injury in I/R [56]. The PINK1/Parkin signaling pathway is the classical pathway for activating mitophagy in mammalian cells. Parkin has a high affinity for phosphorylated S65-ubiquitin and is translocated from the cytoplasm to the mitochondrial surface to promote the ubiquitination of outer mitochondrial membrane (OMM) proteins during mitophagy [57]. Many of the actions of S1P are shared by the structurally related SPC, and these two lipids may thus share common mechanisms and sites of action. Meanwhile, it has been shown that SPC can be hydrolyzed into S1P [58]. Within the scope of this investigation, we have not delineated whether the action of SPC within cardiomyocytes is a consequence of direct receptor engagement or subsequent to its hydrolysis into S1P. This ambiguity must be resolved in subsequent experimental inquiries to unambiguously delineate the molecular mechanism of SPC’s mode of action.

Although NR4A2 is associated with the pathogenesis of various heart diseases, the role of NR4A2 in I/R heart injury is unclear. One study reported that propofol mitigated myocardial H/R injury by inhibiting the expression of miR-449a and consequently upregulating the expression of NR4A2 [27]. This suggested that NR4A2 exerts protective effects on myocardial I/R injury. During myocardial I/R injury, lincRNA-p21 upregulates the expression of NR4A2 through the downregulation of miR-466i-5p, promoting myocardial cell apoptosis in myocardial I/R injury [28]. This study suggested that NR4A2 overexpression exacerbates myocardial I/R injury. Previously, we demonstrated that SPC can alleviate hypoxic myocardial injury by upregulating the expression of NR4A2 [29]. In particular, SPC upregulated NR4A2 expression during the reperfusion phase. NR4A2, a transcription factor, can directly activate the transcription and translation of OPTN. For OPTN to participate in mitophagy as a receptor, OPTN must be recruited to the OMM through its ubiquitin-binding domain, which binds polyubiquitinated substrates and transports them to the autophagosome and subsequently to the lysosome via the LC3 II interaction domain [59]. Parkin is the key factor for the recruitment of OPTN to the mitochondria [60]. This study demonstrated that SPC upregulated OPTN expression via NR4A2 and activated Parkin activity. Although NR4A2 was not involved in Parkin activation, the activated Parkin-mediated recruitment of OPTN increased its co-localization with the mitochondria, promoting the co-localization of LC3 II with the mitochondria and mitophagy [61]. Therefore, SPC activates Parkin and upregulates OPTN expression via NR4A2. This promotes OPTN translocation to the mitochondria, replenishes the number of mitophagy-related receptors, and promotes mitophagy in I/R cardiomyocytes. The molecular mechanisms underlying the myocardial protective effects of SPC involving the inhibition of cardiomyocyte apoptosis are summarized in Figure 10.

## 4. Materials and Methods

### 4.1. Chemicals and Reagents

Rat myocardial cell line (H9c2 cells), human umbilical vein endothelial cells (HUVECs), Henrietta Lacks cells (HeLa cells), and human embryonic kidney 293 cells with SV40 large T antigen (293T) were obtained from the Shanghai Cell Bank of the Chinese Academy of Sciences. Dulbecco’s modified Eagle’s medium (DMEM) was purchased from Gibco (Grand Island, NY, USA). The anti-β-actin (sc-8432), anti-Caspase3 (sc-7148), anti-PARP1 (sc-7150), anti-ATX (sc-374222), anti-NR4A2 (sc-376984), anti-Optineurin (sc-166576), anti-Parkin (sc-30130) primary antibodies, and Staurosporine (sc-3510) were purchased from Santa Cruz Biotechnology (Dallas, TX, USA). COX IV (WL02203) was purchased from Wanleibio (Shenyang, China). Anti-LC3-I/II (#4108S) antibody was purchased from Cell Signaling Technology (Shanghai, China). The Cy3-conjugated goat anti-rabbit IgG (H + L) (AS007) and Cy3-conjugated goat anti-mice IgG (H + L) (AS008) antibodies were purchased from Abclonal (Wuhan, China). Horseradish peroxidase (HRP)-conjugated goat anti-mice (IH-0031) and HRP-conjugated goat anti-rabbit (IH-0011) secondary antibodies were purchased from Dingguo Changsheng Biotechnology Co., Ltd. (Beijing, China). Electrochemiluminescence HRP substrate was obtained from Millipore (Burlington, MA, USA). Mito-tracker Green (KGMP007) and DiO (KGMP003) were purchased from KeyGEN Bio (Nanjing, China).

### 4.2. Cell Culture

H9c2 cells, HeLa cells, 293T cells, and HUVECs were cultured in DMEM or minimal essential medium supplemented with 10% fetal bovine serum (FBS), 2 mM glutamine, 100 U/mL penicillin, and 100 mg/mL streptomycin in humidified air (5% CO_2_) at 37 °C. Transient transfection was performed using Lipofectamine 2000 (Lipo2000; Thermo Fisher Scientific, Inc., Waltham, MA, USA), following the manufacturer’s instructions. Cells were seeded in 6-well or 24-well plates and washed with phosphate-buffered saline (PBS). The cells were then transfected with siRNAs against ATX (NM_001040092.3), NR4A2 (NM_019328.3), and OPTN (NM_145081.4) (Guangzhou RiboBio Co., Ltd., Guangzhou, China) (Table 1) using Lipo2000 at 40–60% confluence for 36 h, following the manufacturer’s instructions. Additionally, the cells were transfected with pcDNA3.1-ATX, pcDNA3.1-NR4A2, and pCMV-OPTN overexpression plasmids using Lipo2000 at 60–80% confluence for 36 h. Cells were then collected for subsequent immunofluorescence staining or Western blotting analysis.

### 4.3. Construction of the Myocardial I/R Injury Mice Model

All C57BL/6 mice used in this study were obtained from the Experimental Animal Center of Weifang Medical College (Weifang, China). Male mice aged 10 weeks were used to construct the myocardial I/R model. Each experimental group consisted of three male mice and three female mice. Briefly, mice were anesthetized using an intraperitoneal injection of 4% pentobarbital sodium (40 mg/kg) [62], intubated with polyethylene-90 tubing, and ventilated with 95% O_2_ and 5% CO_2_ with a rodent ventilator (HX–101E, Chengdu Taimeng Software Co., Ltd., Chengdu, China). The body temperature of mice was maintained between 34 °C and 37 °C on a warm pad. An oblique incision was introduced to expose the left anterior descending coronary artery, which was then ligated with a 6-0 silk suture 1 mm below the tip of the left atrial appendage. After 45 min of coronary occlusion, the ligation was released to allow tissue reperfusion, which was visualized. Next, the chest wall was closed, and the mice were sacrificed after 24 h for TTC staining or sample collection.

Previous experiments were conducted by exogenously injecting SPC into the tail vein once a day for three days [63]. In this study, the same tail vein injection method was used to inject exosomes. A preliminary experiment was performed to determine the optimal injection dosage gradient of exosomes (25 μg/10 g bodyweight) to exert cardioprotective effects.

### 4.4. Tetrazolium Chloride (TTC) Staining

The myocardial tissue samples were collected and immediately frozen at −20 °C for approximately 20 min. The tissue sections with a thickness of 2 mm were then extracted using a scalpel. The samples were subsequently incubated in a solution of 2% TTC at 37 °C in the dark for 20 min. Next, the tissue sections were fixed with 4% paraformaldehyde solution for 24 h and imaged. The images were analyzed using Image-Pro Plus 6.0 [64].

### 4.5. Echocardiography

Cardiac function was assessed by an ultrasound system with a 23-MHz frequency transducer (Vinno 6LAB, Vinno Technology Co., Ltd., Suzhou, China) before surgery and 24 h after surgery. Mice were anesthetized with isoflurane (3%) to achieve heart rates stabilized at about 400 beats/min and hair was removed over the measurement area. The mice were then placed in a supine position on a heating pad. To measure ejection fraction and fractional shortening, short axis images were acquired at the level of the papillary muscle with M-mode. EF and FS were calculated according to the standard formulae [65].

### 4.6. Establishment of the H/R Cell Model

The H9c2 cells were cultured in DMEM supplemented with 10% FBS until 80% confluency. The cell culture medium was replaced with serum-free low-glucose DMEM, and the cells were cultured for 1 h at 95% N_2_ and 5% CO_2_. Next, the medium was replaced with DMEM, and the cells were cultured at 95% air and 5% CO_2_ for 6 h or subjected to reoxygenation. SPC (5 μM) was added at the beginning of reoxygenation [66]. Controls were incubated under normoxic conditions for 12 h [67].

### 4.7. Isolation and Identification of HUVEC-Exos

FBS was centrifuged overnight at 100,000× *g*. The supernatant was collected and used to supplement DMEM at a final concentration of 10%. This medium was used to culture HUVEC for 48 h. The cell culture supernatant was subjected to centrifugation using a high-speed cryo-centrifuge (Thermo Fisher Scientific, Inc., Waltham, MA, USA) at 300× *g* for 5 min. The supernatant was collected and centrifuged again at 2000× *g* for 10 min. Next, the supernatant was collected and centrifuged again at 10,000× *g* for 1 h. This centrifugation was repeated once. Finally, the supernatant was subjected to ultracentrifugation using an ultracentrifuge (Beckman Coulter, Inc., Brea, CA, USA) at 110,000× *g* for 70 min. The supernatant was carefully discarded, and the precipitate at the bottom of the centrifuge tube was resuspended in sterile PBS and centrifuged at 110,000× *g* for 70 min. The supernatant was carefully discarded to obtain the precipitate containing the purified HUVEC-Exos, which were resuspended in PBS and stored at −80 °C for further use.

The isolated HUVEC-Exos were successively washed with PBS, pre-fixed in PBS (pH = 7.4) containing 2.5% glutaraldehyde, fixed in PBS containing 1% osmium tetroxide for 2 h, incubated on glow-discharged copper grids for 1 min, and stained with 2% phosphotungstic acid. The excess buffer was absorbed using filter paper, and the grids were stained with 0.2% uranyl acetate (pH = 7.0) for 40 s. After air-drying at room temperature, the HUVEC-Exos were observed under a transmission electron microscope at 80 keV. The exosomes were identified using nanoparticle tracking analysis. Briefly, the exosomes were diluted to a volume of 1 mL in a trehalose pulse medium. The size and concentration of exosomes were measured using the Nanosight System on ZetaView PMX 110 (Particle Metrix, Inning am Ammersee, Bavaria, Germany). The levels of exosome-specific proteins were measured using Western blotting analysis with the anti-CD63 (1:1000, WL02549) and anti-TSG101 (1:1000, WL05130) antibodies (Wanleibio, Shenyang, Liaoning, China).

### 4.8. Cardiac Lipid Extraction

After the residual blood was removed, the heart tissue was sectioned and transferred to 1.5 mL centrifuge tubes. The weight of the sample was accurately measured. The samples were homogenized with ultrasound on ice. The homogenate (100 μL) was transferred to a new centrifuge tube and extracted with 3 mL of methanol–chloroform mixture (methanol/chloroform = 1:3) for 4 h. This extraction was repeated thrice. The impurities in the extract were removed using a 0.22-μm membrane filter. The filtrate was aliquoted into 10 mL centrifuge tubes and incubated in a refrigerator at −80 °C for at least 8 h. The frozen filtrate was dried overnight with a vacuum freeze dryer for at least 12 h. The sample was dissolved in 200 μL pure methanol.

### 4.9. Purification of SPC Using HPLC

SPC was purified using the HPLC system equipped with an intelligent HPLC pump (Model 880-PU, Jasco, Tokyo, Japan) and an intelligent sampler (Model 851-AS, Jasco, Japan) under the following conditions: column, Develosil ODS-HG-5 (internal diameter = 2.0 mm and length = 35.0 mm; Nomura Chemical, Aichi, Japan); mobile phase A, 5 mM ammonium formate/methanol/tetrahydrofuran (5:2:3) and 0.1% formic acid; mobile phase B, 5 mM ammonium formate/methanol/tetrahydrofuran (1:2:7) with 0.1% formic acid; the stepwise gradients of mobile phase A and mobile phase B were set from 100:0 to 0:100; sample volume, 20 μL; column temperature, 40 °C; flow rate, 0.2 mL/min.

### 4.10. Western Blotting Analysis

Mice myocardial tissues and H9c2 cells were lysed/homogenized in RIPA lysis buffer (radioimmunoprecipitation assay buffer) (Thermo Fisher, Beijing, China) for 30 min at 4 °C. The lysate was centrifuged at 12,000× *g* and 4 °C for 10 min. The mitochondrial and cytoplasmic proteins were extracted using the cell mitochondrial isolation kit (C3601, Beyotime, Shanghai, China). Proteins (25–50 μg/lane) were subjected to sodium dodecyl sulfate–polyacrylamide gel electrophoresis. The resolved proteins were transferred to a membrane. The membrane was incubated with anti-PARP1 (1:2000), anti-cleaved caspase3 (1:2000), anti-Parkin (1:2000), anti-COX IV (1:3000), and anti-β-actin (1:5000) antibodies at 4 °C for 16 h. After washing with Tris-buffered saline containing Tween-20 (TBST), the samples were incubated with HRP-conjugated secondary antibodies (1:5000–1:20,000) at room temperature for 1 h. The immunoreactive signals were developed using HRP substrate and analyzed with ImageJ (V2.9.0, NIH, Bethesda, MD, USA). Immunoblots represent three (in vitro) or six (in vivo) independent experiments.

We prepared each run including two lanes for each experimental group and a positive (pos) and a negative (neg) control to ensure the rigor of our Western blotting results. The positive controls for NR4A2 and β-actin are HeLa whole cell lysate; the positive controls for caspase3 and PARP1 are HeLa (with 1 μM Staurosporine) whole cell lysate; the positive controls for OPTN is 293T whole cell lysate; the positive controls for Parkin is HUVEC whole cell lysate; the positive controls for COX IV is mouse heart lysates. All cellular and tissue lysates are obtained post-lysis with RIPA lysis buffer. The negative control is the RIPA lysis buffer. Proteins (25–50 μg/lane) of the positive controls were subjected to sodium dodecyl sulfate–polyacrylamide gel electrophoresis. The same volume of the RIPA lysis buffer was added to the negative control.

### 4.11. Mitophagy Assay

H9c2 cells were seeded in 24-well plates and incubated with Mito-tracker Green at 37 °C for 90 min [68,69]. The samples were washed with PBS and fixed with 4% paraformaldehyde for 15 min. Mitophagy in cells was observed under a laser confocal microscope (Carl Zeiss lsm900, Carl Zeiss AG, Baden-Württemberg, Germany). The level of mitophagy was analyzed and quantified using ImageJ as previously described [70]. The expression of mitophagy-related factors, such as Parkin was detected using Western blotting to quantify the level of mitophagy in H9c2 cells as previously described [70].

### 4.12. Dual-Luciferase Reporter Assay

The promoter region of *OPTN* was cloned and inserted upstream of the gene encoding luciferase in the pGL3 vector. The recombinant vector was co-transfected into 293T cells with pGL3 Renilla luciferase control plasmid using Lipo2000. Additionally, the cells were transfected with the pcDNA3.1-NR4A2 overexpression vector. Renilla luciferase was used as a normalization reference. At 48 h post-transfection, the 293T cells were lysed and subjected to luciferase activity measurement with the dual-luciferase assay kit (Beyotime, Shanghai, China), following the manufacturer’s instructions. Firefly and Renilla luciferase activities were determined using a plate reader. The activity of firefly luciferase was normalized to that of Renilla luciferase [71].

### 4.13. Statistical Analysis

Data from at least three independent experiments are expressed as mean ± standard error of mean (SEM). Means were compared using Student’s *t*-test or one-way analysis of variance (ANOVA). We verified the normal distribution of the values intended for the Student’s *t*-test using the Kolmogorov–Smirnov Test. Differences between the groups were determined using one-way ANOVA with Tukey post hoc test. All statistical analyses were performed using GraphPad Prism 8 (GraphPad Software, La Jolla, CA, USA). Differences were considered significant at *p* < 0.05 (* *p* < 0.05, ** *p* < 0.01, *** *p* < 0.0005, and **** *p* < 0.0001).

## 5. Conclusions

This study demonstrated that SPC-rich VEC-Exos protect cardiomyocytes against I/R-induced cellular injury. Additionally, this study elucidated the molecular mechanisms underlying the protective effects of SPC against I/R-induced cardiomyocyte injury. SPC inhibited cardiomyocyte apoptosis by promoting mitophagy through the activation of the Parkin pathway and the upregulation of OPTN expression in an NR4A2-dependent manner. These findings suggest that endogenous SPC in VEC-Exos is a novel regulator of mitophagy and apoptosis in cardiomyocytes. Thus, SPC in VEC-Exos is a potential therapeutic target for I/R cardiac injury.

## Figures and Tables

**Figure 1 ijms-25-03305-f001:**
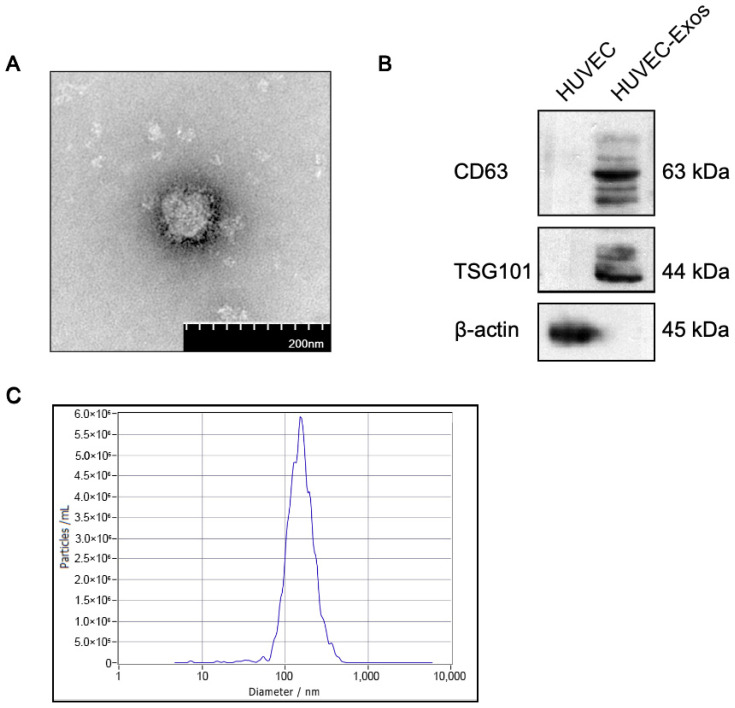
Characterization of human umbilical vascular endothelial cell-derived exosomes (HUVEC-Exos). (**A**) Transmission electron microscopy analysis of HUVEC-Exos. Scale bar = 200 nm. (**B**) Western blotting analysis of the expression of the exosome markers CD63 and TSG101 in HUVEC-Exos. (**C**) Analysis of the concentration and diameter of exosomes using nanoparticle tracking analysis.

**Figure 2 ijms-25-03305-f002:**
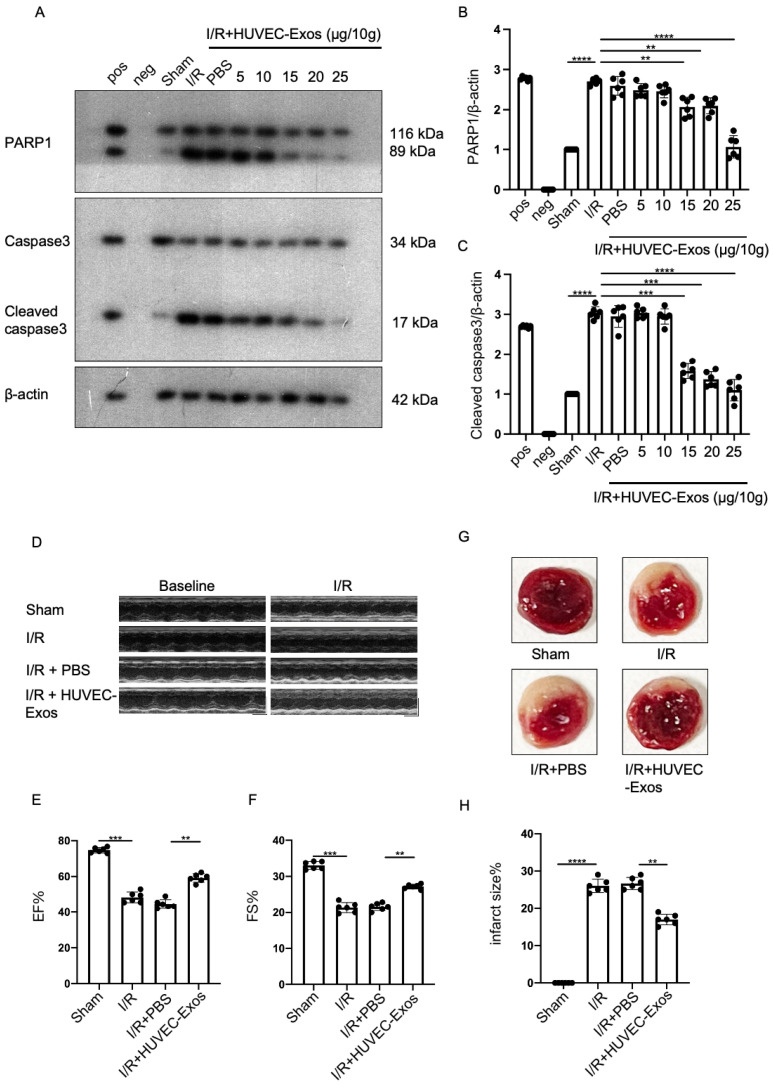
HUVEC-Exos mitigate myocardial ischemia/reperfusion (I/R) injury in mice. (**A**–**C**) Effect of tail vein exosome administration on the myocardial expression of apoptotic markers (PARP1 and cleaved caspase3) was evaluated using western blotting analysis (*n* = 6 for each group). Pos: HeLa (with 1 μM Staurosporine) whole cell lysate; neg: RIPA lysis buffer. (**D**) Echocardiographic assessments of cardiac functions (*n* = 6 for each group). (**E**) Ejection fraction (EF) of mice (*n* = 6 for each group). (**F**) Shortening fraction (FS) of mice (*n* = 6 for each group). (**G**,**H**) Tetrazolium chloride (TTC) staining was performed to examine the myocardial infarct area (*n* = 6 for each group). In all statistical graphs, values are mean ± S.E.M. ** *p* < 0.01, *** *p* < 0.001, and **** *p* < 0.0001. All data are representative of six independent experiments.

**Figure 3 ijms-25-03305-f003:**
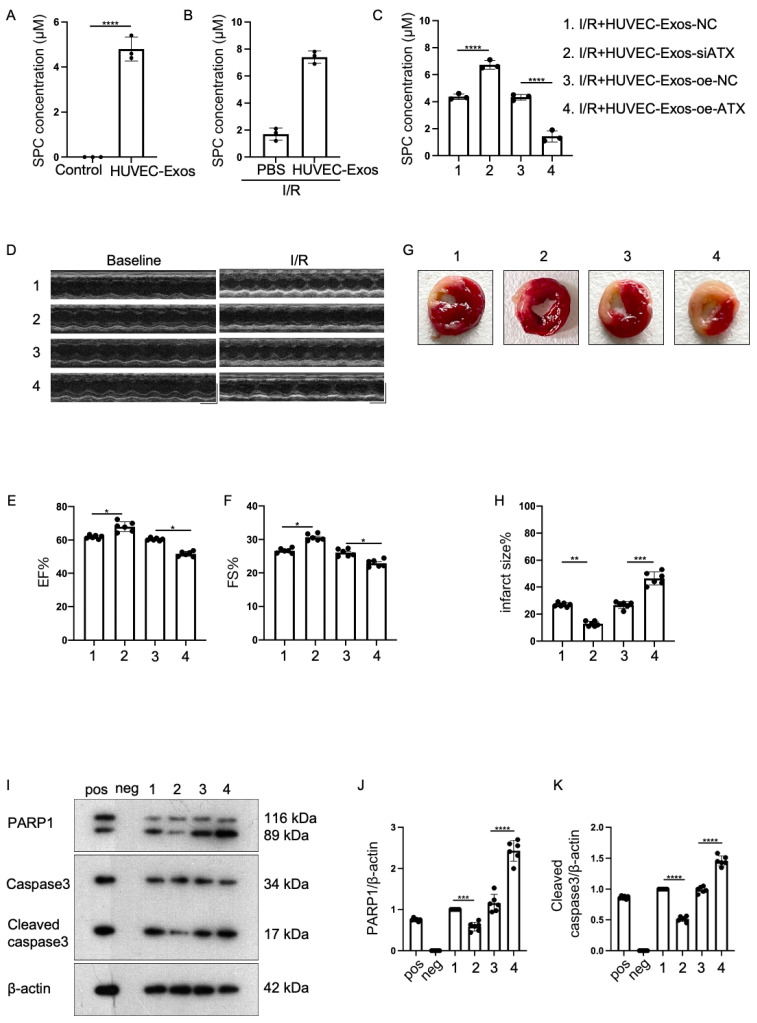
Cardioprotective role of sphingosylphosphorylcholine (SPC) in HUVEC-Exos in mice. (**A**) SPC in HUVEC-Exos was detected using high-performance liquid chromatography equipped with an evaporative light scattering detector (HPLC-ELSD) (*n* = 3 for each group). (**B**) Effect of HUVEC-Exos on the mice myocardial SPC content was examined using HPLC-ELSD (*n* = 3 for each group). (**C**) Effect of si-ATX-transfected or pcDNA3.1-ATX-transfected HUVEC-Exos on the mice myocardial SPC content was examined using HPLC-ELSD (*n* = 3 for each group). (**D**) Echocardiographic assessments of cardiac functions (*n* = 6 for each group). (**E**) Ejection fraction (*n* = 6 for each group). (**F**) Shortening fraction (*n* = 6 for each group). (**G**,**H**) Effect of si-ATX-transfected or pcDNA3.1-ATX-transfected HUVEC-Exos on the infarct area was examined using tetrazolium chloride (TTC) staining (*n* = 6 for each group). (**I**–**K**) Western blotting analysis of the myocardial levels of apoptotic proteins (PARP1 and cleaved caspase3) (*n* = 6 for each group). Pos: HeLa (with 1 μM Staurosporine) whole cell lysate; neg: RIPA lysis buffer. In all statistical graphs, values are mean ± S.E.M. * *p* < 0.05, ** *p* < 0.01, *** *p* < 0.001, and **** *p* < 0.0001. All data are representative of six independent experiments.

**Figure 4 ijms-25-03305-f004:**
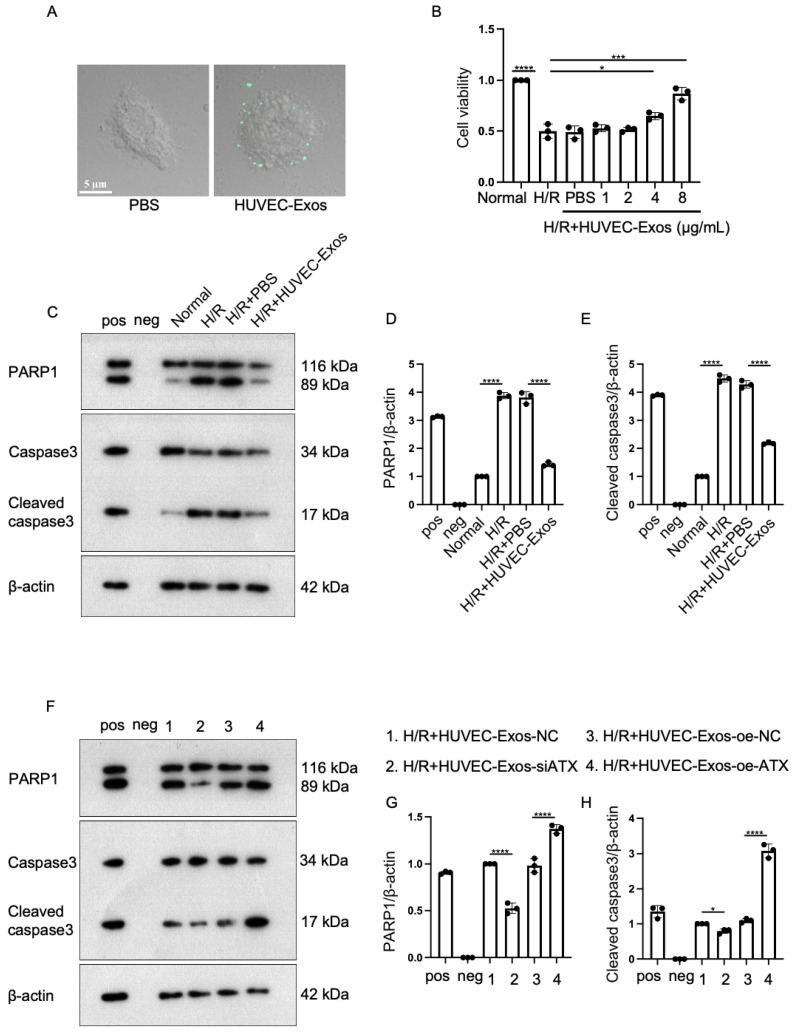
In vitro validation of the function of SPC in HUVEC-Exos using H9c2 cells. (**A**) Laser confocal microscopy revealed the internalization of HUVEC-Exos by H9c2 cells. Scale bar = 5 μm (**B**) Cardiomyocyte viability was examined using the 3-(4,5-dimethylthiazol-2-yl)-2,5-diphenyltetrazolium bromide assay (*n* = 3 for each group). (**C**–**E**) Western blotting analysis of the effect of HUVEC-Exos on the expression of apoptosis markers (PARP1 and cleaved caspase3) in H/R-treated H9c2 cells (*n* = 3 for each group). Pos: HeLa (with 1 μM Staurosporine) whole cell lysate; neg: RIPA lysis buffer. (**F**–**H**) Western blotting analysis of the expression levels of apoptosis-related proteins (PARP1 and cleaved caspase3) in H9c2 cells (*n* = 3 for each group). Pos: HeLa (with 1μM Staurosporine) whole cell lysate; neg: RIPA lysis buffer. In all statistical graphs, values are mean ± S.E.M. * *p* < 0.05, *** *p* < 0.001, and **** *p* < 0.0001. All data are representative of three independent experiments.

**Figure 5 ijms-25-03305-f005:**
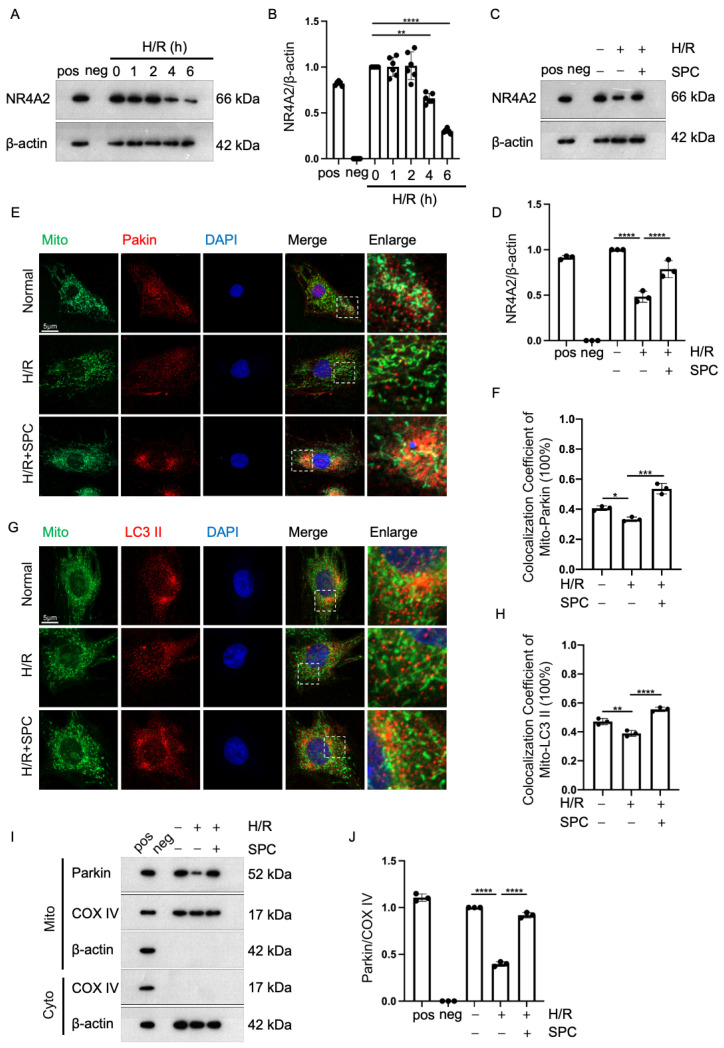
SPC promotes mitophagy in H9c2 cells through NR4A2. (**A**,**B**) Western blotting analysis of the NR4A2 levels in H9c2 cells after reoxygenation for 1, 2, 4, and 6 h (*n* = 3 for each group). Pos: HeLa whole cell lysate; neg: RIPA lysis buffer. (**C**,**D)** The effect of SPC on the NR4A2 level in H/R-treated H9c2 cells (*n* = 3 for each group). Pos: HeLa whole cell lysate; neg: RIPA lysis buffer. (**E**,**F**) Laser confocal microscopy analysis of the effect of SPC on the co-localization of Parkin and LC3 II with the mitochondria in H/R-treated H9c2 cells. Green fluorescence indicates mitochondria, red fluorescence indicates Parkin and LC3 II, and blue fluorescence indicates the nucleus (*n* = 3 for each group). Scale bar = 5 μm. (**G**,**H**) The co-localization of Parkin and LC3 II with the mitochondria was analyzed using ImageJ (V2.9.0) (*n* = 3 for each group). Scale bar = 5 μm. (**I**,**J**) Effect of SPC on the mitochondrial Parkin levels in H/R-induced H9c2 cells (*n* = 3 for each group). Pos: mix of HUVEC whole cell lysate, mouse heart lysates, and HeLa whole cell lysate; neg: RIPA lysis buffer. The white boxes in the immunofluorescence images indicate enlarged areas. In all statistical graphs, values are mean ± S.E.M. * *p* < 0.05, ** *p* < 0.01, *** *p* < 0.001, and **** *p* < 0.0001. All data are representative of three independent experiments.

**Figure 6 ijms-25-03305-f006:**
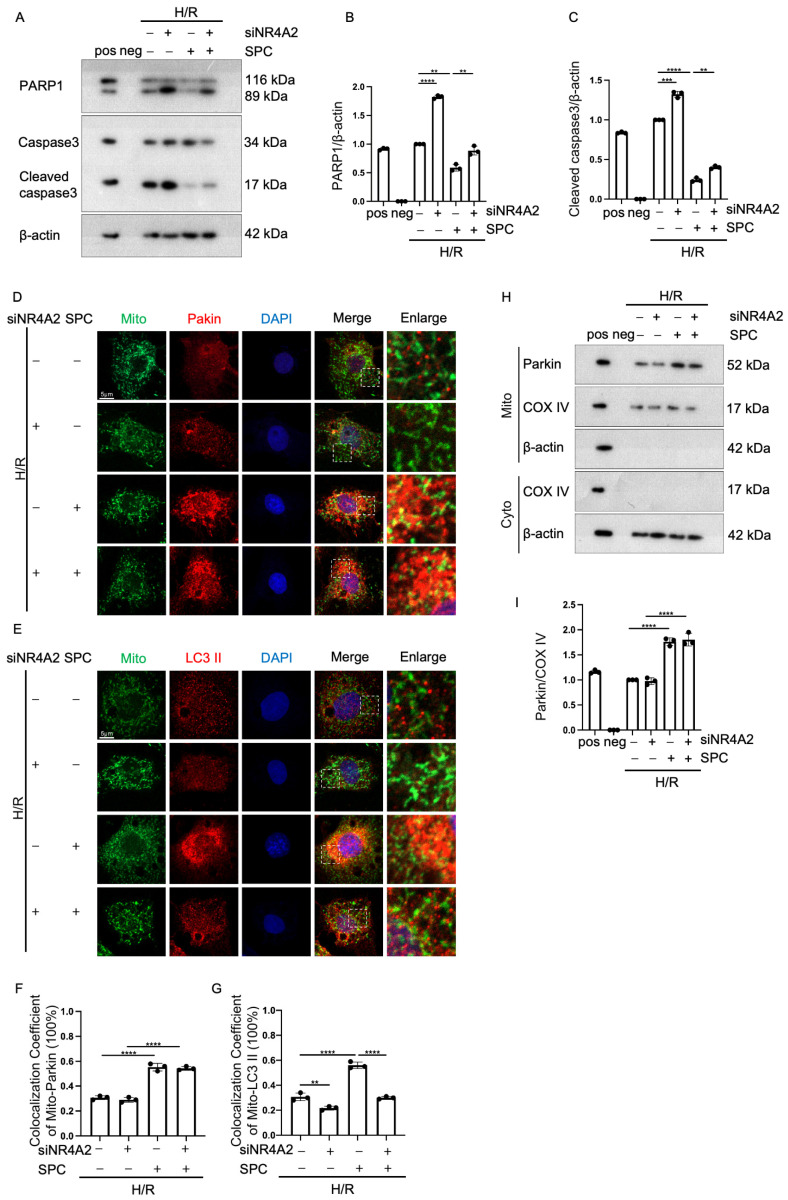
*NR4A2* knockdown mitigates the protective effect of SPC in H9c2 cells. (**A**–**C**) Western blotting analysis revealed that the expression of apoptotic proteins (PARP1 and cleaved caspase3) was upregulated in NR4A2 knockdown H9c2 cells (*n* = 3 for each group). Pos: HeLa (with 1 μM Staurosporine) whole cell lysate; neg: RIPA lysis buffer. (**D**–**E**) Laser confocal microscopy analysis of the effect of NR4A2 knockdown on the co-localization of Parkin and LC3 II with the mitochondria in H/R-treated H9c2 cells. Green fluorescence indicates mitochondria, red fluorescence indicates Parkin and LC3 II, and blue fluorescence indicates the nucleus (*n* = 3 for each group). Scale bar = 5 μm. (**F**–**G**) The co-localization of Parkin and LC3 II with the mitochondria was analyzed using ImageJ (V2.9.0) (*n* = 3 for each group). (**H**–**I**) Western blotting analysis of the mitochondrial expression level of Parkin in NR4A2 knockdown cells (*n* = 3 for each group). Pos: mix of HUVEC whole cell lysate, mouse heart lysates, and HeLa whole cell lysate; neg: RIPA lysis buffer. The white boxes in the immunofluorescence images indicate enlarged areas. In all statistical graphs, values are mean ± S.E.M. ** *p* < 0.01, *** *p* < 0.001, and **** *p* < 0.0001. All data are representative of three independent experiments.

**Figure 7 ijms-25-03305-f007:**
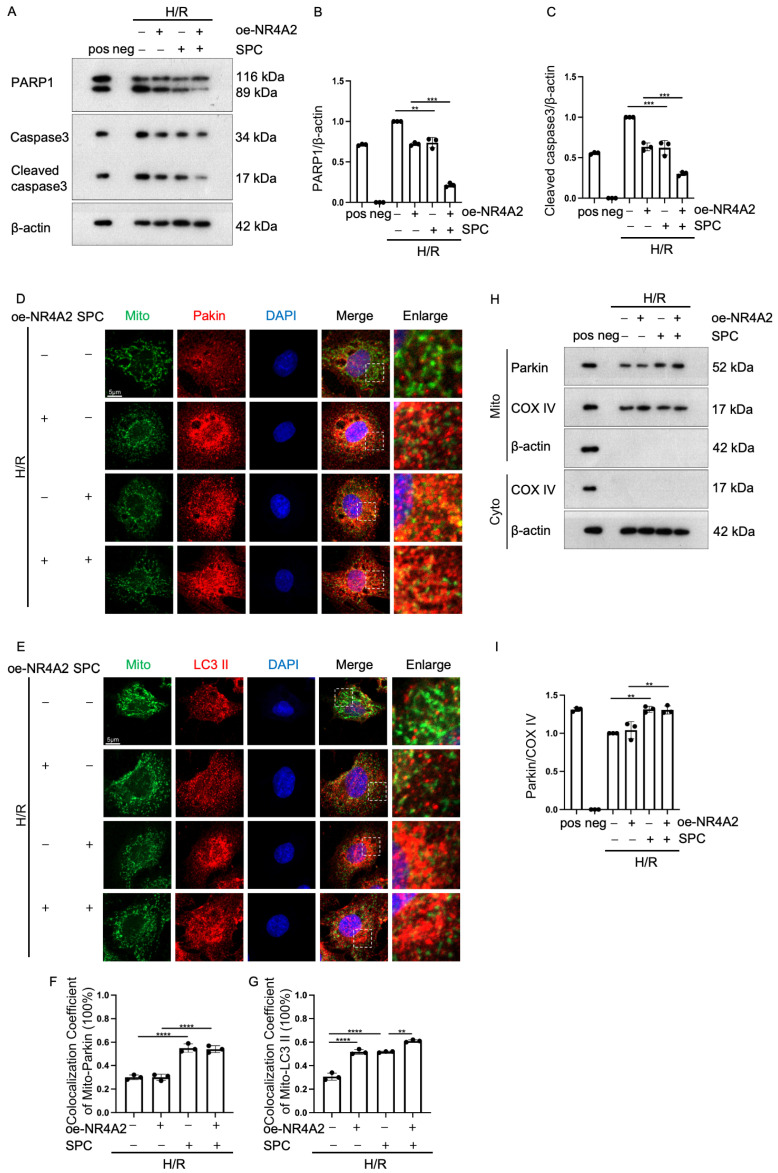
NR4A2 overexpression potentiates the protective effects of SPC in H9c2 cells. (**A**–**C**) Western blotting analysis of the expression of apoptotic proteins (PARP1 and cleaved caspase3) in NR4A2-overexpressing H9c2 cells (*n* = 3 for each group). Pos: HeLa (with 1 μM Staurosporine) whole cell lysate; neg: RIPA lysis buffer. (**D**,**E**) Laser confocal microscopy analysis of the effect of NR4A2 overexpression on the co-localization of Parkin and LC3 II with the mitochondria in H/R-treated H9c2 cells. Green fluorescence indicates mitochondria, red fluorescence indicates Parkin and LC3 II, and blue fluorescence indicates the nucleus (*n* = 3 for each group). Scale bar = 5 μm. (**F**,**G**) The co-localization of Parkin and LC3 II with the mitochondria was analyzed using ImageJ (V2.9.0) (*n* = 3 for each group). (**H**,**I**) Western blotting analysis of the mitochondrial expression levels of Parkin in NR4A2-overexpressing H9c2 cells (*n* = 3 for each group). Pos: mix of HUVEC whole cell lysate, mouse heart lysates, and HeLa whole cell lysate; neg: RIPA lysis buffer. The white boxes in the immunofluorescence images indicate enlarged areas. In all statistical graphs, values are mean ± S.E.M. ** *p* < 0.01, *** *p* < 0.001, and **** *p* < 0.0001. All data are representative of three independent experiments.

**Figure 8 ijms-25-03305-f008:**
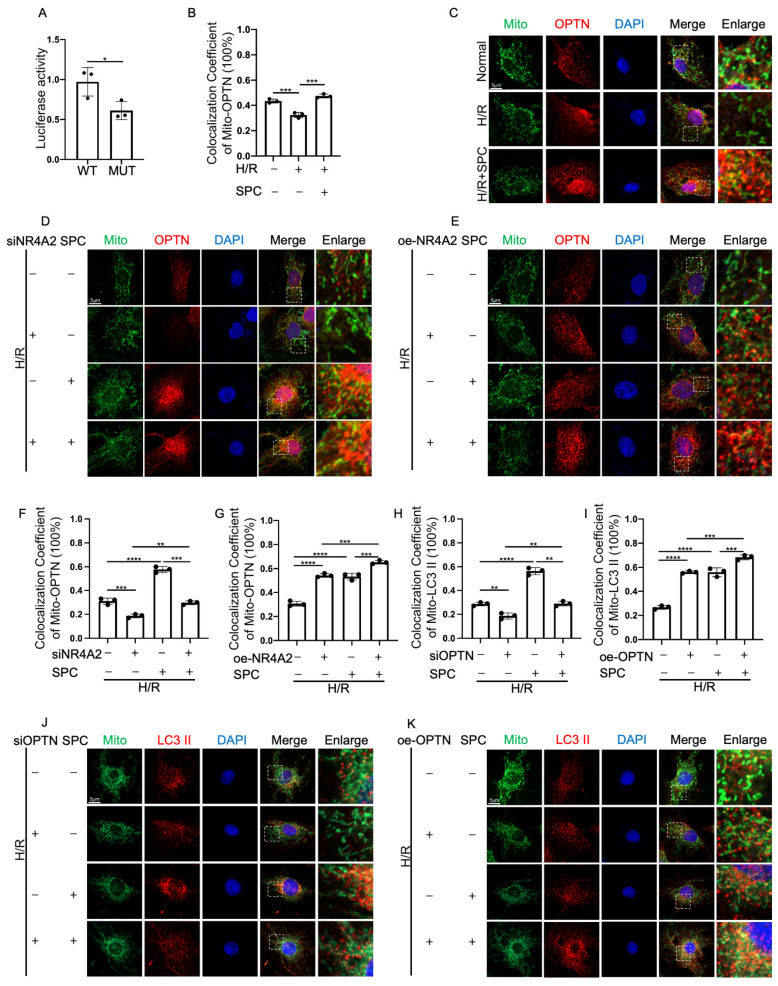
NR4A2 mediates mitophagy in H9c2 cells through OPTN. (**A**) Luciferase activity in 293T cells was evaluated using the dual-luciferase reporter gene assay (*n* = 3 for each group). (**B**,**C**) Laser confocal microscopy analysis of the effect of SPC on the co-localization of OPTN with the mitochondria in H/R-treated H9c2 cells. The co-localization of OPTN with the mitochondria was analyzed using ImageJ (V2.9.0). Green fluorescence indicates mitochondria, red fluorescence indicates LC3 II, and blue fluorescence indicates the nucleus (*n* = 3 for each group). Scale bar = 5 μm. (**D**–**G**) Laser confocal microscopy analysis of the effect of OPTN knockdown or OPTN overexpression on the co-localization of OPTN with the mitochondria in H/R-treated H9c2 cells. The co-localization of OPTN with the mitochondria was analyzed using ImageJ (V2.9.0) (*n* = 3 for each group). Scale bar = 5 μm. (**H**–**K**) Laser confocal microscopy analysis of the effect of OPTN knockdown or OPTN overexpression on the co-localization of LC3 II with the mitochondria in H/R-treated H9c2 cells. The co-localization of LC3 II with the mitochondria was analyzed using ImageJ (V2.9.0) (*n* = 3 for each group). Scale bar = 5 μm. The white boxes in the immunofluorescence images indicate enlarged areas. In all statistical graphs, values are mean ± S.E.M. * *p* < 0.05, ** *p* < 0.01, *** *p* < 0.001, and **** *p* < 0.0001. All data are representative of three independent experiments.

**Figure 9 ijms-25-03305-f009:**
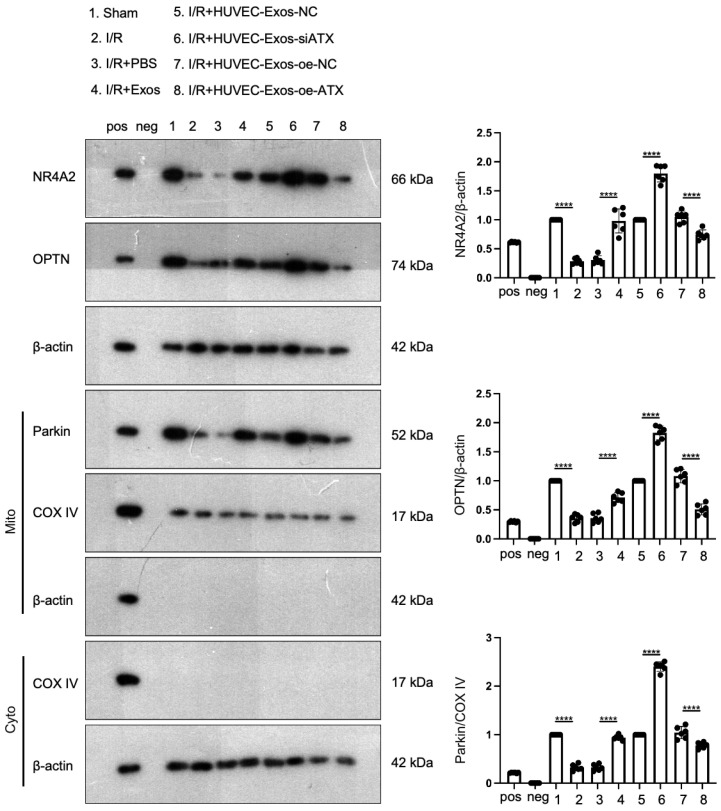
Differential expression levels of NR4A2, OPTN, and Parkin in the myocardial tissues of I/R mice administered with HUVEC-Exos exhibiting differential SPC contents. Western blotting analysis of the effect of exosomes on the myocardial expression levels of NR4A2, OPTN, and Parkin (*n* = 6 for each group). Pos: mix of HUVEC whole cell lysate, 293T whole cell lysate, mouse heart lysates, and HeLa whole cell lysate; neg: RIPA lysis buffer. In all statistical graphs, values are mean ± S.E.M. **** *p* < 0.0001. All data are representative of six independent experiments.

**Figure 10 ijms-25-03305-f010:**
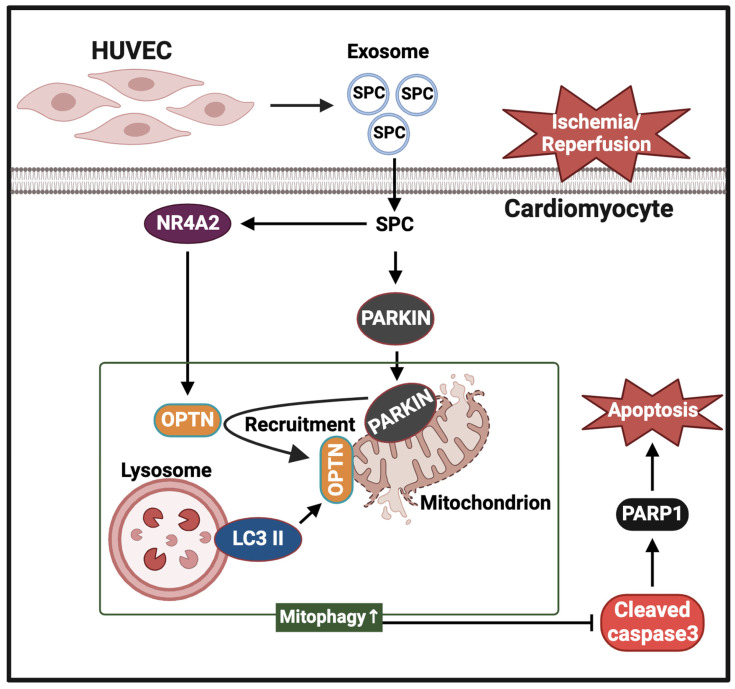
The molecular mechanisms underlying the inhibitory effects of SPC on I/R-induced apoptosis in cardiomyocytes. SPC-rich vascular endothelial cell-derived exosomes are accessible to cardiomyocytes. SPC inhibits I/R-induced cardiomyocyte apoptosis by activating the Parkin and NR4A2/OPTN pathways and promoting mitochondrial autophagy.

**Table 1 ijms-25-03305-t001:** SiRNA primers and their sequences used in this study. “-F” and “-R” indicate the forward and reverse primers, respectively.

Primer	Sequence (5′ → 3′)
siATX-F	AUCGACAAAAUUGUGGGGCTT
siATX-R	GCCCCACAAUUUUGUCGAUTT
NC-ATX-F	UUCUCCGAACGUGUCACGUTT
NC-ATX-R	ACGUGACACGUUCGGAGAATT
siNR4A2-F	AAGCGCCGCCGAAAUCGUUGU
siNR4A2-R	ACAACGAUUUCGGCGGCGCUU
NC-NR4A2-F	UAUCGGAACCCUAGGUUCCTT
NC-NR4A2-R	GGAACCUAGGGUUCCGAUATT
siOPTN-F	GGAAACACUGAGCAUUCAATT
siOPTN-R	UUGAAUGCUCAGUGUUUCCTT
NC-OPTN-F	UUCUCCGAACGUGUCACGUTT
NC-OPTN-R	ACGUGACACGUUCGGAGAATT

## Data Availability

All data generated and analyzed during this study were included in this published article.

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
