# Peer review of "Vascular Endothelial Cell-Derived Exosomal Sphingosylphosphorylcholine Attenuates Myocardial Ischemia–Reperfusion Injury through NR4A2-Mediated Mitophagy"

_ijms, 2024, doi:10.3390/ijms25063305_

Round 1

Reviewer 1 Report (New Reviewer)

Comments and Suggestions for Authors

Reviewer report

Vascular Endothelial Cell-derived Exosomal Sphingosylphosphorylcholine Attenuates Myocardial Ischemia-Reperfusion Injury Through NR4A2-mediated Mitophagy. 

This is a well-organized study showing the importance of Sphingosylphosphorylcholine which highlights lights the potential therapeutic effects of SPC-rich exosomes secreted by VECs on inhibiting Ischemia-Reperfusion Injury in cardiomyocytes and in animal model

A.    Why few sections in the articles are highlighted please correct it.

B.    Which gender mice were used in the study whether male or female please explain and justify the selected gender.

C.    Isoflurane is administered as an inhalation anesthetic agent author has mentioned isoflurane was administered by the intraperitoneal route please add the reference citing intraperitoneal route of Isoflurane.

D.    Section 4.5 Color Doppler echocardiography analysis of mice is incomplete please add a full description of the methodology including transducer and frequency used with the minimum heart rate maintained.

E.    Please mention whether data was captured in Brightness mode or not because Color Doppler doesn’t give data in M mode for EF and FS.

F.     What will be the effect on wall thickness, systolic and diastolic volume please explain.

Author Response

Vascular Endothelial Cell-derived Exosomal Sphingosylphosphorylcholine Attenuates Myocardial I/R Injury Through NR4A2-mediated Mitophagy. 

This is a well-organized study showing the importance of Sphingosylphosphorylcholine which highlights lights the potential therapeutic effects of SPC-rich exosomes secreted by VECs on inhibiting I/R Injury in cardiomyocytes and in animal model

A. Why few sections in the articles are highlighted please correct it.

Response: Thank you for your feedback. Our manuscript was resubmitted after revision. During the previous submission, the academic editor provided some comments and suggested that we clearly mark the revised sections and resubmit the revised manuscript, which is why certain parts of the document were highlighted. We've removed all the previous highlighting, and the highlighting we're keeping now is what's been reworked in this manuscript.

B. Which gender mice were used in the study whether male or female please explain and justify the selected gender.

Response: Thank you for your perceptive observations. In our study, each experimental group consisted of 3 male mice and 3 female mice. The reason for selecting both genders is to mitigate any potential effects of sex differences on the experimental outcomes, ensuring the reliability and general applicability of our results. We have clarified this issue in the Materials and Methods section of the manuscript, on page 21, from line 475 to line 476. The content is as follows:

“Each experimental group consisted of 3 male mice and 3 female mice.”

C. Isoflurane is administered as an inhalation anesthetic agent author has mentioned isoflurane was administered by the intraperitoneal route please add the reference citing intraperitoneal route of Isoflurane.

Response: Thank you for providing your valuable feedback. We apologize for the error. Prior to the I/R procedure, we anesthetized the mice via intraperitoneal injection of sodium pentobarbital; whereas for cardiac function assessments before and after surgery, the mice were anesthetized with inhaled isoflurane during echocardiographic examination. Due to an oversight, these two anesthesia methods were confused in the Materials and Methods section. We have corrected this error on lines 498 to 499 of the Materials and Methods section in the manuscript. The corrected content is as follows:

“Mice were anesthetized using an intraperitoneal injection of 4% pentobarbital sodium (40 mg/kg) [1].”

D. Section 4.5 Color Doppler echocardiography analysis of mice is incomplete please add a full description of the methodology including transducer and frequency used with the minimum heart rate maintained.

Response: Thank you for your thoughtful comments. In the Materials and Methods section, lines 520 to 526, we have provided a complete description of the echocardiographic methodology used for cardiac assessment, including transducer and frequency used with the minimum heart rate maintained. The content is as follows:

“Cardiac function was assessed by an ultrasound system with a 23-MHz frequency transducer (Vinno 6LAB, Vinno Technology) before surgery and 24 h after surgery。Mice were anesthetized with isoflurane (3%) to achieve heart rates stabilized at about 400 beats/min and hair was removed over the measurement area. The mice were then placed in a supine position on a heating pad. To measure ejection fraction and fractional shortening, short axis images were acquired at the level of the papillary muscle with M-mode. EF, and FS were calculated according to the standard formulae[2].”

E. Please mention whether data was captured in Brightness mode or not because Color Doppler doesn’t give data in M mode for EF and FS.

Response: Thank you for your meticulous review and for calling this matter to our attention. I would like to clarify that our intention was to describe the overall structural and functional assessment of the heart, for which the technique "echocardiography" should have been applied rather than " Color Doppler echocardiography ". We have corrected this issue in line 519 of the Materials and Methods section. The content is as follows:

“4.5. Echocardiography”

In addition, after collecting the cardiac ultrasound, we can analyze the collected images, we randomly select four consecutive cardiac cycle targets and mark the diastolic ventricular wall thickness and systolic ventricular wall thickness sequentially, subsequently the values of EF and FS will be obtained. This procedure is done automatically by our cardiac ultrasound machine.

F. What will be the effect on wall thickness, systolic and diastolic volume please explain.

Response: Thank you for your attentive review and suggestions. In the aftermath of a cardiac I/R injury, there is a complex cascade of physiological changes that profoundly affect ventricular wall dynamics and chamber volumes. During the acute phase post-injury, the affected myocardial region often undergoes an increase in wall thickness, primarily attributed to cellular swelling (edema), infiltration of inflammatory cells, and interstitial hemorrhage. As the injury progresses into the subacute phase, the wall may begin to thin out due to the loss of viable myocardium and replacement by fibrotic tissue, which lacks contractile function[3].

Functionally, systolic volume is frequently reduced in the setting of I/R injury. This is primarily because the ischemic insult leads to myocardial stunning or hibernation, resulting in impaired myocardial contractility and systolic dysfunction. Consequently, the heart's ability to eject blood effectively during systole is compromised, manifesting as a decreased stroke volume and ejection fraction.Conversely, diastolic volume often increases after I/R injury, as a result of ventricular remodeling, where the heart attempts to compensate for the loss of contractile function by dilating. This ventricular dilation is accompanied by impaired ventricular relaxation (diastolic dysfunction), which impedes proper filling of the heart and exacerbates the increase in diastolic volume[4].

These structural and functional adaptations can be quantified using echocardiography or magnetic resonance imaging (MRI) and are pivotal endpoints in clinical and experimental studies. The resulting systolic and diastolic dysfunction contribute to a decline in cardiac output and can precipitate heart failure if left unchecked[5]. We hope that this response addresses the effect of cardiac I/R injury on wall thickness, as well as systolic and diastolic volumes. Thank you for the opportunity to clarify these important aspects of our study.

References:

  1. Li, Y., et al., microRNA-130a-5p suppresses myocardial ischemia reperfusion injury by downregulating the HMGB2/NF-kappaB axis. BMC Cardiovasc Disord, 2021. 21(1): p. 121.
  2. Wang, X., et al., Cardiac disruption of SDHAF4-mediated mitochondrial complex II assembly promotes dilated cardiomyopathy. Nat Commun, 2022. 13(1): p. 3947.
  3. Algoet, M., et al., Myocardial ischemia-reperfusion injury and the influence of inflammation. Trends Cardiovasc Med, 2023. 33(6): p. 357-366.
  4. Diakos, N.A., et al., Afterload-induced left ventricular diastolic dysfunction during myocardial ischaemia and reperfusion. Exp Physiol, 2015. 100(3): p. 288-301.
  5. Ibanez, B., et al., Evolving therapies for myocardial ischemia/reperfusion injury. J Am Coll Cardiol, 2015. 65(14): p. 1454-71.

Reviewer 2 Report (Previous Reviewer 3)

Comments and Suggestions for Authors

Thank authors for the revisions, which have significantly improved the manuscript. I am satisfied with the explanations and modifications made by the authors. The addition of in vivo experiments has enriched the experiment content, while standardizing the nomenclature in the figures has improved readability. 

My only minor comment is that there are many repetitive phrases in the manuscript. For example, in Figure 2 legend, it is unnecessary to add the explanation of "Means were compared using one-way ANOVA" after each statistical graph. Additionally, the statement "In addition to the experimental groups, we also added positive and negative control groups to ensure the rigor of our western blotting results" could be included in the methods section instead of being repeated in each conclusion. This would streamline the manuscript and make it more concise.

Author Response

Thank authors for the revisions, which have significantly improved the manuscript. I am satisfied with the explanations and modifications made by the authors. The addition of in vivo experiments has enriched the experiment content, while standardizing the nomenclature in the figures has improved readability. 

My only minor comment is that there are many repetitive phrases in the manuscript. For example, in Figure 2 legend, it is unnecessary to add the explanation of "Means were compared using one-way ANOVA" after each statistical graph. Additionally, the statement "In addition to the experimental groups, we also added positive and negative control groups to ensure the rigor of our western blotting results" could be included in the methods section instead of being repeated in each conclusion. This would streamline the manuscript and make it more concise.

Response: Thank you for your constructive comments. I have carefully reviewed the manuscript and removed the repetitive phrases to improve the clarity and conciseness of the text. These changes have streamlined the manuscript considerably. I appreciate your attention to detail and believe that these revisions have enhanced the overall quality of the manuscript.

Reviewer 3 Report (Previous Reviewer 2)

Comments and Suggestions for Authors

The manuscript # ijms-2887312-peer-review-v1 has been abundantly edited, however, I could not find the sentence where the authors clarify the use of SPC monomers and why they did that: “We have previously demonstrated that HUVEC-Exos can exert robust cardioprotective effects via SPC. Next, we aim to elucidate the molecular mechanisms by which SPC functions. Considering the complexity of exosomal contents and the potential for other substances to interfere with experimental outcomes, we will proceed to treat H9c2 cells with SPC monomers to validate the molecular mechanisms underlying SPC’s effects.” The authors specified that it should have been clarified in their manuscript from line 180 on page 9 to line 184 on page 10 but this is not the case.

Comments on the Quality of English Language

Has been significantly improved.

Author Response

The manuscript # ijms-2887312-peer-review-v1 has been abundantly edited, however, I could not find the sentence where the authors clarify the use of SPC monomers and why they did that: “We have previously demonstrated that HUVEC-Exos can exert robust cardioprotective effects via SPC. Next, we aim to elucidate the molecular mechanisms by which SPC functions. Considering the complexity of exosomal contents and the potential for other substances to interfere with experimental outcomes, we will proceed to treat H9c2 cells with SPC monomers to validate the molecular mechanisms underlying SPC’s effects.” The authors specified that it should have been clarified in their manuscript from line 180 on page 9 to line 184 on page 10 but this is not the case.

Response: Thank you for your diligence in reviewing our manuscript. Upon revisiting the document, we indeed find that the intended clarification regarding the use of SPC monomers was mistakenly omitted from the previous submission.

As per your valuable feedback, we have now incorporated the necessary explanation into the manuscript from lines 199 to 203 on page 9. The content is as follows:

“We have previously demonstrated that HUVEC-Exos can exert robust cardioprotective effects via SPC. Next, we aim to elucidate the molecular mechanisms by which SPC functions. Considering the complexity of exosomal contents and the potential for other substances to interfere with experimental outcomes, we will proceed to treat H9c2 cells with SPC monomers to validate the molecular mechanisms underlying SPC’s effects.” Thank you for the opportunity to clarify these important aspects of our study.

This manuscript is a resubmission of an earlier submission. The following is a list of the peer review reports and author responses from that submission.

Round 1

Reviewer 1 Report

Comments and Suggestions for Authors

SUMMARY

     The authors provide a clear, well-organized and rigorously interpreted demonstration of how exosome-mediated delivery of sphingo-sylphosphorylcholine (SPC) to cardiomyocytes diminished the severity of cardiac damage caused by ischemia/reperfusion (I/R) in a mouse model.    The investigators further show that SPC attenuates cardiomyocyte depletion through apoptosis, which is a major mechanism of I/R injury to the heart; and activates the Parkin and nuclear receptor subfamily group A member 2/optineurin (NR4A2/OPTN) pathways.    These outcomes improve understanding of underlying processes of I/R pathogenesis.  Moreover, this study is particularly impressive since it demonstrates a promising strategy for modulating mitophagy levels in future efforts to prevent or treat cardiovascular disease.  The topic material of this report is timely and addresses a major medical need.  It is anticipated that it will stimulate reader interest and also serve as a motivator for expanded investigation of exosomes, with cargoes of SPC and related metabolites.  The investigators tested an easily understood hypothesis in drug design – and produced data which was interpreted conservatively.     This reviewer here conditionally recommends publication if the authors are willing to make revisions in the present manuscript according to the comments below:

REVIEWER COMMENTS

(1) Use of the English language was adequate to communicate the substance of this report and was mostly grammatically correct.   However throughout the body of the manuscript phraseology was used that was somewhat awkward and would benefit by polishing by a native speaker of English with experience in scientific writing.    It is strongly(!) suggested that the authors arrange for a working review of the manuscript by some individual – perhaps a colleague, with fluency in both written and spoken English and experience in scientific writing.  This individual should identify all sections of the paper which contain awkward or imprecise wording and rephrase these sections using elegant grammar.

(2)  The authors should instruct the scientific writer selected for review and linguistic adjustment to compare the original Abstract with the rewritten version included here at the bottom of this review.   This rewritten Abstract is provided for the authors as a model of the phraseology which should be used throughout the fully revised manuscript.  The authors are welcome to splice this new Abstract (below) into their revised manuscript verbatim.

(3)  The authors should insert a table containing full-length names for specialized terminology and the corresponding abbreviations used in the text.

REWRITTEN ABSTRACT

NOTE:  This section was revised by a peer reviewer for this manuscript and provided to the authors as an example of how the remainder of the text may be presented to readers with optimum clarity and scientific accuracy.  The authors are welcome to splice this Abstract into the revised article verbatim as shown below.  Moreover the writer they select for linguistic refinement of this paper, as advised in comment 2 above, should use the style of phraseology used in this rewritten Abstract (below) as a guide for improvement of the remainder of the document.

ABSTRACT

Cardiomyocyte survival is a critical contributing processs of host adaptive responses to cardiovascular diseases (CVD). Cells of the cardiovascular endothelium have recently been reported to promote cardiomyocyte survival through exosome-loading cargos. Sphingo-sylphosphorylcholine (SPC), an intermediate metabolite of sphingolipids, mediates protection against myocardial infarction (MI).   Nevertheless, the mechanism of SPC delivery by vascular endothelial cell (VEC)-derived exosomes (VEC-Exos) remain uncharacterized at the time of this writing.  The present study utilized a mouse model of ischemia/reperfusion (I/R) to demonstrate that administration of exosomes via tail vein injection significantly diminished the severity of I/R-induced cardiac damage and prevented apoptosis of cardiomyocytes.   Moreover, SPC was here identified as the primary mediator of the observed protective effects of VEC-Exos.   Also within this investigation, in vitro experiments using cardiomyocytes showed that SPC counteracted myocardial I/R injury by activating the Parkin and nuclear receptor subfamily group A member 2/optineurin (NR4A2/OPTN) pathways, in turn resulting in increased levels of mitophagy within I/R-affected myocardium.   The present study highlights the potential therapeutic effects of SPC-rich exosomes secreted by VECS on alleviating I/R-induced apoptosis in cardiomyocytes, thereby providing strong experimental evidence to support the application of SPC as a potential therapeutic target in prevention and treatment of myocardial infarction.

RECOMMENDATION:  This report represents an enormous and dedicated commitment to exploration of an area of vital importance in a wide range of ongoing lines of biomedical research and clinical practice.     If the authors are willing to modify the manuscript according to the critiques above, this reviewer recommends publication.

Comments on the Quality of English Language

This is an excellent paper.    Make the linguistic refinements and it will be superlative.    

Author Response

SUMMARY

     The authors provide a clear, well-organized and rigorously interpreted demonstration of how exosome-mediated delivery of sphingo-sylphosphorylcholine (SPC) to cardiomyocytes diminished the severity of cardiac damage caused by ischemia/reperfusion (I/R) in a mouse model.    The investigators further show that SPC attenuates cardiomyocyte depletion through apoptosis, which is a major mechanism of I/R injury to the heart; and activates the Parkin and nuclear receptor subfamily group A member 2/optineurin (NR4A2/OPTN) pathways.    These outcomes improve understanding of underlying processes of I/R pathogenesis.  Moreover, this study is particularly impressive since it demonstrates a promising strategy for modulating mitophagy levels in future efforts to prevent or treat cardiovascular disease.  The topic material of this report is timely and addresses a major medical need.  It is anticipated that it will stimulate reader interest and also serve as a motivator for expanded investigation of exosomes, with cargoes of SPC and related metabolites.  The investigators tested an easily understood hypothesis in drug design – and produced data which was interpreted conservatively.     This reviewer here conditionally recommends publication if the authors are willing to make revisions in the present manuscript according to the comments below:

REVIEWER COMMENTS

(1) Use of the English language was adequate to communicate the substance of this report and was mostly grammatically correct. However throughout the body of the manuscript phraseology was used that was somewhat awkward and would benefit by polishing by a native speaker of English with experience in scientific writing.    It is strongly(!) suggested that the authors arrange for a working review of the manuscript by some individual – perhaps a colleague, with fluency in both written and spoken English and experience in scientific writing.  This individual should identify all sections of the paper which contain awkward or imprecise wording and rephrase these sections using elegant grammar.

Response: Thanks for your valuable and thoughtful comments. As requested, we have now proofread the entire manuscript by a professor of biology in the USA to improve the overall presentation of the language of our revised manuscript without changing any of our original significance of our study. We hope that we have now revised our manuscript to your satisfaction. We are willing to revise our manuscript upon your further request. The mailing address of the professor who helped us polish the manuscript is as follows:

Prof. Fengjie Sun

Department of Biological SciencesSchool of Science and Technology

Georgia Gwinnett College

1000 University Center Lane

Lawrenceville, GA 30043, USA

(2)  The authors should instruct the scientific writer selected for review and linguistic adjustment to compare the original Abstract with the rewritten version included here at the bottom of this review.   This rewritten Abstract is provided for the authors as a model of the phraseology which should be used throughout the fully revised manuscript.  The authors are welcome to splice this new Abstract (below) into their revised manuscript verbatim.

Response: Thank you for giving your input and making the changes. We splice this new Abstract into our revised manuscript verbatim. Additionally, we have revised the remaining sections of the manuscript to align with the style of phraseology used in this rewritten Abstract.

(3)  The authors should insert a table containing full-length names for specialized terminology and the corresponding abbreviations used in the text.

Response: Thanks for your suggestion. On the page 1 of the manuscript, line 25, we have inserted a table containing full-length names for specialized terminology and the corresponding abbreviations used in the text. The inserted table is as follows:

Abbreviations

Abbreviation

Full name

ATX

autotaxin

H/R

hypoxia/reoxygenation

H9c2

rat cardiomyoblasts cell line

HUVEC

human umbilical vein endothelial cells

I/R

ischemia/reperfusion

LVEF

left ventricular ejection fraction

LVFS

left ventricular fraction shortening

NR4A2

nuclear receptor subfamily 4 group A member 2

OPTN

optineurin

SPC

sphingosylphosphorylcholine

TEM

transmission electron microscopy

REWRITTEN ABSTRACT

NOTE:  This section was revised by a peer reviewer for this manuscript and provided to the authors as an example of how the remainder of the text may be presented to readers with optimum clarity and scientific accuracy.  The authors are welcome to splice this Abstract into the revised article verbatim as shown below.  Moreover the writer they select for linguistic refinement of this paper, as advised in comment 2 above, should use the style of phraseology used in this rewritten Abstract (below) as a guide for improvement of the remainder of the document.

ABSTRACT

Cardiomyocyte survival is a critical contributing processs of host adaptive responses to cardiovascular diseases (CVD). Cells of the cardiovascular endothelium have recently been reported to promote cardiomyocyte survival through exosome-loading cargos. Sphingo-sylphosphorylcholine (SPC), an intermediate metabolite of sphingolipids, mediates protection against myocardial infarction (MI).   Nevertheless, the mechanism of SPC delivery by vascular endothelial cell (VEC)-derived exosomes (VEC-Exos) remain uncharacterized at the time of this writing.  The present study utilized a mouse model of ischemia/reperfusion (I/R) to demonstrate that administration of exosomes via tail vein injection significantly diminished the severity of I/R-induced cardiac damage and prevented apoptosis of cardiomyocytes.   Moreover, SPC was here identified as the primary mediator of the observed protective effects of VEC-Exos. Also within this investigation, in vitro experiments using cardiomyocytes showed that SPC counteracted myocardial I/R injury by activating the Parkin and nuclear receptor subfamily group A member 2/optineurin (NR4A2/OPTN) pathways, in turn resulting in increased levels of mitophagy within I/R-affected myocardium.   The present study highlights the potential therapeutic effects of SPC-rich exosomes secreted by VECS on alleviating I/R-induced apoptosis in cardiomyocytes, thereby providing strong experimental evidence to support the application of SPC as a potential therapeutic target in prevention and treatment of myocardial infarction.

RECOMMENDATION:  This report represents an enormous and dedicated commitment to exploration of an area of vital importance in a wide range of ongoing lines of biomedical research and clinical practice.     If the authors are willing to modify the manuscript according to the critiques above, this reviewer recommends publication.

Reviewer 2 Report

Comments and Suggestions for Authors

This paper describes the impact of sphingosylphosphorylcholine (SPC) in exosomes produced by HUVEC cells on attenuation of MI-reperfusion injury in a mouse model and describes mechanistic responses through NR4A2-mediated mitophagy. The manuscript needs a bit of polishing but is fairly well written. It describes nicely designed experiments to study the in depth the mechanism through which SPC can act in the H9c2 cells. However, in its current state, I kind of feel there are some gaps in the paper that limits its impact.

One major comment regards the need to use exosomes to get the benefits om MI-reperfusion injury and the originality of the paper. Although it is not really clearly described in the Material and Methods nor in the Results section, it appears that most in vitro experiments with H9c2 cells were using free SPC (or SPC dissolved in BSA?) as treatments instead of exosomes containing SPC.  Please clarify. If that is correct, then, why use exosomes to treat the animals? It would be much simpler to simply use directly pure SPC, (or SPC adsorbed on BSA), no? There might be some advantages of SPC incorporated into exosomes vs free SPC or SPC, but there is no comparison in the reported experiments to conclude in the benefits of using exosomes over simply SPC/BSA. Interestingly, it appears that such experiments with SPC/BSA were performed in 2010 (PMID: 21274265). A comparison of exosomes to SPC/BSA would be appropriate to conclude on the therapeutic potential of the exosomes reported here.

On the other hand, one may say that the paper kind of described an endogenous mechanism of protection to MI. Obviously, this mechanism fails in some individuals. A question may be, why? For instance, what could alter the SPC content of endothelial cells in relation to atherosclerosis processes? Does exposure of HUVECs to LDL decrease SPC content in exosomes (or their production by ECs?)? Would inflammation result in decrease in SPC content of the exosomes? You could expose the cells to TNFalpha for instance and see what happens. Another point of view may be to study SPC in patients’ samples. Do they have less SPC? Less exosomes? Less SPC in exosomes? Etc… In relation to the animal model, Is it possible that the level of SPC in the animals differ after MI? What about the exosomes and their SPC content? Is SPC mainly associated to exosomes or to HDL or else?

Minor comments:

In the result section, line 100, you mention that MI was induced for 45 minutes, however in the MM section, line 398 it says 30 minutes of occlusion. Please clarify.

What is the concentration of SPC used for the mitophagy assay and other in vitro experiments?

The discussion could gain some value if there is some knowledge to be gained about ATX regulation in relation to CVD.

Comments on the Quality of English Language

The English is fair and can be improved.

Author Response

This paper describes the impact of sphingosylphosphorylcholine (SPC) in exosomes produced by HUVEC cells on attenuation of MI-reperfusion injury in a mouse model and describes mechanistic responses through NR4A2-mediated mitophagy. The manuscript needs a bit of polishing but is fairly well written. It describes nicely designed experiments to study the in depth the mechanism through which SPC can act in the H9c2 cells. However, in its current state, I kind of feel there are some gaps in the paper that limits its impact.

Response: Thanks for your valuable and thoughtful comments. As requested, we have now proofread the entire manuscript by a professor of biology in the USA to improve the overall presentation of the language of our revised manuscript without changing any of our original significance of our study. We hope that we have now revised our manuscript to your satisfaction. We are willing to revise our manuscript upon your further request. The mailing address of the professor who helped us polish the manuscript is as follows:

Prof. Fengjie Sun

Department of Biological SciencesSchool of Science and Technology

Georgia Gwinnett College

1000 University Center Lane

Lawrenceville, GA 30043, USA

  1. One major comment regards the need to use exosomes to get the benefits om MI-reperfusion injury and the originality of the paper. Although it is not really clearly described in the Material and Methods nor in the Results section, it appears that most in vitro experiments with H9c2 cells were using free SPC (or SPC dissolved in BSA?) as treatments instead of exosomes containing SPC.  Please clarify. If that is correct, then, why use exosomes to treat the animals? It would be much simpler to simply use directly pure SPC, (or SPC adsorbed on BSA), no? There might be some advantages of SPC incorporated into exosomes vs free SPC or SPC, but there is no comparison in the reported experiments to conclude in the benefits of using exosomes over simply SPC/BSA. Interestingly, it appears that such experiments with SPC/BSA were performed in 2010 (PMID: 21274265). A comparison of exosomes to SPC/BSA would be appropriate to conclude on the therapeutic potential of the exosomes reported here.

Response: Thank you for your professional suggestions. We focused our research on the cardioprotective effects of SPC delivered by exosomes, considering their innate capability for substance transport, inherent prolonged circulation, and excellent biocompatibility, making them promising drug delivery vehicles that could potentially prevent the premature degradation of free SPC in the bloodstream. In Fig. 3A of our manuscript, we quantified the SPC content in exosomes, using approximately 1μg of exosomes per sample, which allowed us to calculate the SPC content as 4μM/μg. In Fig. 2A, our findings indicated that an exosome dosage of 15μg/10g/day (1.5μg/g/day) already exhibited protective effects, corresponding to an SPC dose of 8μM/g/day. The exosome dosage used in this experiment was 25μg/10g/day, which translates to an SPC dose of 10μM/g/day and demonstrated even stronger cardioprotective effects. In previously published research from our lab, we studied the protection offered by SPC monomers to mice with myocardial infarction, with the effective dose being 10μM/g/day. Our preliminary analysis suggests that the effective SPC dosage when using exosomes for cardioprotection might be lower than that of the SPC monomers. However, since we did not directly compare the two under identical experimental conditions, our analysis remains preliminary and cannot yield definitive results. Therefore, your remark, "A comparison of exosomes to SPC/BSA would be appropriate to conclude on the therapeutic potential of the exosomes reported here," is incredibly helpful. It points out the limitations of our current experiment and offers new directions for future research. We plan to further modify exosomes by altering their membrane composition to increase their targeting to damaged cardiac regions, and then compare the therapeutic effects of free SPC, unmodified exosomes, and modified exosomes.

  1. On the other hand, one may say that the paper kind of described an endogenous mechanism of protection to MI. Obviously, this mechanism fails in some individuals. A question may be, why? For instance, what could alter the SPC content of endothelial cells in relation to atherosclerosis processes? Does exposure of HUVECs to LDL decrease SPC content in exosomes (or their production by ECs?)? Would inflammation result in decrease in SPC content of the exosomes? You could expose the cells to TNFalpha for instance and see what happens.

Response: Thank you for raising these meaningful questions, which greatly assist us in refining our experiments and designing future ones. First off, the reason for this mechanism failure in some individuals might be that during myocardial ischemia-reperfusion injury, SPC levels drop below normal, impeding its protective efficacy. We have diligently reviewed the paper you cited earlier (PMID: 21274265) and noted the intricate association between SPC and HDL. Moreover, your reminder alerted us to the fact that during myocardial I/R injury, LDL, particularly oxidized LDL (oxLDL), can initiate an inflammatory response, attracting inflammatory cells to the affected area, and prompting the release of cytokines and chemotactic agents. LDL also contributes to the reduction of SPC levels. Hence, LDL or the inflammation it induces could be one of the factors leading to the decrease in SPC levels, an aspect we had previously disregarded. At present, due to the unavailability of the required experimental materials and reagents, we are unable to swiftly conduct additional experiments to confirm your suggestions. However, we are planning to undertake further experiments where we will expose the cells to TNFalpha for instance and see what happens.

  1. Another point of view may be to study SPC in patients’ samples. Do they have less SPC? Less exosomes? Less SPC in exosomes? Etc…

Response: Thanks for your constructive comment. The current research does not clarify the changes in SPC levels in myocardial infarction patients. We have not assessed whether the SPC levels, exosome secretion, or SPC content within exosomes are altered in these patients. In subsequent experiments, we will attempt to detect changes in relevant indicators within samples from myocardial infarction patients.

  1. In relation to the animal model, Is it possible that the level of SPC in the animals differ after MI? What about the exosomes and their SPC content? Is SPC mainly associated to exosomes or to HDL or else?

Response: Thanks for your constructive comment. In our subsequent experiments, we found that after I/R injury, the SPC levels in mouse serum were significantly lower than normal. Furthermore, in our in vitro studies where we cultured endothelial cells and induced I/R injury, the exosomes secreted showed notably reduced levels of SPC compared to normal. Current research advancements and our findings do not conclusively determine the relationship between SPC, exosomes or HDL. However, our further results suggest that changes in SPC levels may be attributable to the β subunit of Acid Ceramidase (AC β). A study published in 2020 (PMID: 33233706) has identified AC β as a critical regulator of SPC production. We hypothesize that during myocardial I/R, dysfunction or abnormal expression of the AC β may hinder normal SPC synthesis, leading to reduced levels. In our later experiments, we discovered that following I/R injury, the functionality of the AC β in endothelial cells is indeed compromised, failing to maintain normal SPC production levels. We are currently investigating the specific causes of AC β subunit dysfunction. Prompted by your reminder, we are preparing to add new experiments to verify whether LDL and HDL also play significant roles in SPC level fluctuations during myocardial I/R injury.

We hope our responses and the revisions made to the manuscript meet your satisfaction. Should you have any concerns, please do not hesitate to contact us. We are willing to make further amendments based on your suggestions.

Minor comments:

  1. In the result section, line 100, you mention that MI was induced for 45 minutes, however in the MM section, line 398 it says 30 minutes of occlusion. Please clarify.

Response: Thank you for pointing out the mistake. The experimental conditions we used entailed 45 minutes of ischemia. The mention of 30 minutes on line 398 in the MM section was an oversight on my part, and I apologize for this error in the manuscript.

  1. What is the concentration of SPC used for the mitophagy assay and other in vitro experiments?

Response: Thanks for your suggestion. Our team has previously verified that SPC at a concentration of 5μM significantly protects cardiomyocytes in vitro. Therefore, this concentration was used in the current experiment. Modifications and additions have been made to the Materials and Methods section line 467 and the reference has been included. The supplementary content is as follows:

“SPC (5 μM) was added at the beginning of reoxygenation[1, 2].”

  1. The discussion could gain some value if there is some knowledge to be gained about ATX regulation in relation to CVD.

Response: Thanks for your constructive comment. Your suggestion has contributed to a more comprehensive and meaningful discussion. Our summarized research progress has been incorporated into the discussion from lines 333-343. The content is as follows:

“ATX is a secreted enzyme with phosphodiesterase and lysophospholipase D activity that produces physiologically active lipophospholipids, particularly lysophosphatidylin (LPA)[3]. In the field of cardiovascular disease, the ATX-LPA signaling axis is considered an important regulator[4]. Studies have shown that ATX activity may be closely related to the development of cardiovascular diseases such as atherosclerosis[5], hypertension[6], heart failure, and myocardial infarction[7], etc. Several experimental models and clinical studies have indicated that inhibition of ATX activity or blockade of ATX-generated LPA signaling reduces atherosclerotic plaque formation, decreases cardiac remodeling and fibrosis, and thus has therapeutic potential for the treatment of cardiovascular disease[8]. Furthermore, research indicates that ATX functions as a lyase, hydrolyzing SPC [9].”

References:

  1. Li, Y., et al., Sphingosylphosphorylcholine alleviates hypoxia-caused apoptosis in cardiac myofibroblasts via CaM/p38/STAT3 pathway. Apoptosis, 2020. 25(11-12): p. 853-863.
  2. Yao, Y., et al., MicroRNA-155-5p/EPAS1/interleukin 6 pathway participated in the protection function of sphingosylphosphorylcholine to ischemic cardiomyocytes. Life Sci, 2021. 264: p. 118692.
  3. Perrakis, A. and W.H. Moolenaar, Autotaxin: structure-function and signaling. J Lipid Res, 2014. 55(6): p. 1010-8.
  4. Salgado-Polo, F., et al., Autotaxin facilitates selective LPA receptor signaling. Cell Chem Biol, 2023. 30(1): p. 69-84 e14.
  5. Zhao, Y., et al., Targeting the autotaxin - Lysophosphatidic acid receptor axis in cardiovascular diseases. Biochem Pharmacol, 2019. 164: p. 74-81.
  6. Chen, Y., et al., Astaxanthin Attenuates Hypertensive Vascular Remodeling by Protecting Vascular Smooth Muscle Cells from Oxidative Stress-Induced Mitochondrial Dysfunction. Oxid Med Cell Longev, 2020. 2020: p. 4629189.
  7. Tripathi, H., et al., Autotaxin inhibition reduces cardiac inflammation and mitigates adverse cardiac remodeling after myocardial infarction. J Mol Cell Cardiol, 2020. 149: p. 95-114.
  8. Araki, T., et al., Serum autotaxin as a novel prognostic marker in patients with non-ischaemic dilated cardiomyopathy. ESC Heart Fail, 2022. 9(2): p. 1304-1313.
  9. Bourgoin, S.G. and C. Zhao, Autotaxin and lysophospholipids in rheumatoid arthritis. Curr Opin Investig Drugs, 2010. 11(5): p. 515-26.

Reviewer 3 Report

Comments and Suggestions for Authors

This manuscript presents innovative findings, demonstrating that HUVECs can generate exosomes rich in SPC. SPC, through the PARKIN and NR4A2/OPTN pathways, promotes mitophagy, thereby alleviating cardiac ischemia-reperfusion injury. The article holds significant implications, with robust and reliable existing data, making it an outstanding manuscript. However, there is room for minor improvements from the following perspectives.

1. The manuscript lacks validation evidence after the transfection of siRNA or plasmids. Please validate the knockdown or overexpression efficiency after transfection with siRNA or plasmids.

2. Figure 3B and C both showed the I/R+HUVEC-EXOs group, but the SPC concentrations differ significantly. In Figure 3B, it is approximately 7 μM, while in Figure 3C, it is around 4 μM, lacking consistency. Please provide an explanation for this discrepancy.

3. Starting from Figure 5C, SPC is directly added to H9C2 cells instead of using HUVEC-EXOs. However, it seems that the concentration of SPC is not annotated in the figure legend or Methods part. 

4. It is expected to see the authors statistically compare colocoliazation of mito-OPTN in Figure 8 F/G between the siNR4A2 group and siNR4A2+SPC group. It seems that the siNR4A2+SPC group is indeed higher than the siNR4A2 group, which can demonstrate that SPC promotes OPTN recruitment through the PARKIN pathway.

5. It would be beneficial to include in vivo validation with animals in the final part of the article. For example, observing different levels or location of NR4A2 or mitophagy-related proteins in I/R mouse heart tissue with injection of different Exos.

Author Response

This manuscript presents innovative findings, demonstrating that HUVECs can generate exosomes rich in SPC. SPC, through the PARKIN and NR4A2/OPTN pathways, promotes mitophagy, thereby alleviating cardiac ischemia-reperfusion injury. The article holds significant implications, with robust and reliable existing data, making it an outstanding manuscript. However, there is room for minor improvements from the following perspectives.”

  1. The manuscript lacks validation evidence after the transfection of siRNA or plasmids. Please validate the knockdown or overexpression efficiency after transfection with siRNA or plasmids.

Response: Thanks for your suggestion. The knockdown or overexpression efficiency after transfection with siRNA or plasmids has been added to the results section as supplementary material.

  1. Figure 3B and C both showed the I/R+HUVEC-EXOs group, but the SPC concentrations differ significantly. In Figure 3B, it is approximately 7 μM, while in Figure 3C, it is around 4 μM, lacking consistency. Please provide an explanation for this discrepancy.

Response: Thanks for your constructive comment. In Figure 3B, we measured the changes in SPC content in the myocardial tissue of mice after injection of exosomes. In Figure 3C, we detected the SPC levels in the exosomes secreted by HUVECs transfected with ATX siRNA and overexpression plasmids. The lack of consistency is due to the differences in the samples and quantities tested.

  1. Starting from Figure 5C, SPC is directly added to H9C2 cells instead of using HUVEC-EXOs. However, it seems that the concentration of SPC is not annotated in the figure legend or Methods part. 

Response: Thank you for pointing out the mistake. Our team has previously verified that SPC at a concentration of 5μM significantly protects cardiomyocytes in vitro. Therefore, this concentration was used in the current experiment. Modifications and additions have been made to the Materials and Methods section line 469 and the reference has been included. The supplementary content is as follows:

“SPC (5 μM) was added at the beginning of reoxygenation[1, 2].”

  1. It is expected to see the authors statistically compare colocalization of mito-OPTN in Figure 8 F/G between the siNR4A2 group and siNR4A2+SPC group. It seems that the siNR4A2+SPC group is indeed higher than the siNR4A2 group, which can demonstrate that SPC promotes OPTN recruitment through the PARKIN pathway.

Response: Thank you for your valuable suggestions, which have made our analysis of the results more comprehensive. we statistically compare colocalization of mito-OPTN in Figure 8 F/G between the siNR4A2 group and siNR4A2+SPC group and we added difference markers to the Figure 8 F/G. The results show that the siNR4A2+SPC group is indeed higher than the siNR4A2 group. Our revisions are as follows:

“Concurrently, we statistically compare colocalization of mito-OPTN between the siNR4A2 group and siNR4A2+SPC group. It seems that the siNR4A2+SPC group is indeed higher than the siNR4A2 group (Figure 8F, G), which can demonstrate that SPC promotes OPTN recruitment through the Parkin pathway.”

  1. It would be beneficial to include in vivo validation with animals in the final part of the article. For example, observing different levels or location of NR4A2 or mitophagy-related proteins in I/R mouse heart tissue with injection of different Exos.

Response: Thanks for your constructive comment. Following your advice, we added data on the changes in NR4A2, OPTN, and Parkin protein levels in the heart tissue of I/R mice after injection with different Exos. The results are consistent with the trends observed in our in vitro experiments. The specific experimental outcomes have been compiled in Results section 2.9:

“2.9. SPC in HUVEC-Exos similarly promotes the expression of NR4A2, OPTN and Parkin in myocardial tissues of I/R mice

In our previous experiments, we found the mechanism by which SPC exerts a protective effect in vitro through the SPC we used. To further verify this result, we injected several exosomes (HUVEC-Exos, HUVEC-Exos-siATX, HUVEC-Exos-oe-ATX) with different contents of SPC into the tail vein of I/R mice respectively, and then detected the changes of NR4A2 and OPTN expression in the myocardium. changes in the expression of NR4A2 and OPTN in the myocardium, and Western blotting results showed that the expression of NR4A2 and OPTN in the myocardium was significantly increased after injection of HUVEC-Exos; mitochondria in the myocardium were isolated and obtained by using the kit, and mitochondrial proteins were extracted, and the Western blotting results showed that the level of Mito- Parkin levels were also significantly elevated; expression of NR4A2, OPTN, and Mito-Parkin was further elevated in the myocardium after injection of HUVEC-Exos-siATX; after injection of HUVEC-Exos-oe-ATX, compared with the subgroup injected with HUVEC-Exos-oe-ATX, the expression of NR4A2, OPTN and Mito-Parkin expression was significantly lower (Figure 9A). The trends presented in these results are consistent with previous in vitro experiments, and thus we confirm that SPC exerts its cardioprotective effects by simultaneously activating Parkin and the NR4A2/OPTN pathway.

References:

  1. Li, Y., et al., Sphingosylphosphorylcholine alleviates hypoxia-caused apoptosis in cardiac myofibroblasts via CaM/p38/STAT3 pathway. Apoptosis, 2020. 25(11-12): p. 853-863.
  2. Yao, Y., et al., MicroRNA-155-5p/EPAS1/interleukin 6 pathway participated in the protection function of sphingosylphosphorylcholine to ischemic cardiomyocytes. Life Sci, 2021. 264: p. 118692.

Round 2

Reviewer 2 Report

Comments and Suggestions for Authors

Regarding my initial comments to clarify whether free SPC or exosomes were used in experiments presented in some figures (Fig. 5, 6, 7 and 8), the authors somewhat clarified the fact that exosomes were used throughout all experiments, but this was not clarified in the paper. Why was the nomenclature changed in those figures? Given that the authors report that monomer (free) SPC could also be used, this must be clarified in the text of the manuscript, and ideally, in the figures themselves or at the minimum, in the legends it should be clearly stated that the authors used exosomes for which the SPC content was quantified.

Comments on the Quality of English Language

Appropriate.

Author Response

Regarding my initial comments to clarify whether free SPC or exosomes were used in experiments presented in some figures (Fig. 5, 6, 7 and 8), the authors somewhat clarified the fact that exosomes were used throughout all experiments, but this was not clarified in the paper. Why was the nomenclature changed in those figures? Given that the authors report that monomer (free) SPC could also be used, this must be clarified in the text of the manuscript, and ideally, in the figures themselves or at the minimum, in the legends it should be clearly stated that the authors used exosomes for which the SPC content was quantified.

Response: Thank you for your professional suggestions. We apologize for not clarifying the issue fully in our previous response. In the experiments corresponding to Figures 1, 2, 3, and 4, we utilized exosomes, as previously explained, to determine if the exosomes secreted by vascular endothelial cells could exert cardioprotective effects through the carried SPC. In contrast, for the experiments associated with Figures 5, 6, 7, and 8, we employed SPC monomers to investigate the molecular mechanism underlying SPC's protective function. The use of SPC monomers, rather than exosomes, was due to the complexity of exosomal contents, which could contain other substances interfering with the verification of SPC's mechanism. Accordingly, the captions "HUVEC-Exos" were used for results related to Figures 1, 2, 3, and 4, and "SPC" for those related to Figures 5, 6, 7, and 8. This point has been clarified in our manuscript from line 180 on page 9 to line 184 on page 10. The content is as follows:

“We have previously demonstrated that HUVEC-Exos can exert robust cardioprotective effects via SPC. Next, we aim to elucidate the molecular mechanisms by which SPC functions. Considering the complexity of exosomal contents and the potential for other substances to interfere with experimental outcomes, we will proceed to treat H9c2 cells with SPC monomers to validate the molecular mechanisms underlying SPC’s effects.”

Thank you for the opportunity to further revise our manuscript, and we apologize again for not addressing your query clearly last time. Should you still find our response lacking clarity, we sincerely hope you will once again share your valuable feedback to enhance the quality of our manuscript.
